# Appraisal of the Genus *Pleurastrum* (Chlorophyta) Based on Molecular and Climate Data

Katia Sciuto [1,*,†], Marion A. Wolf [2,†], Michele Mistri [1] and Isabella Moro [3]

1 Department of Chemical, Pharmaceutical and Agricultural Sciences, University of Ferrara, Via Luigi Borsari 46, 44121 Ferrara, Italy
2 Department of Environmental Sciences, Informatics and Statistics, Ca' Foscari University of Venice, Via Torino 155, 30172 Mestre, Italy
3 Department of Biology, University of Padova, Via Ugo Bassi 58/B, 35131 Padova, Italy
* Correspondence: katia.sciuto@unife.it
† These authors contributed equally to this work.

**Abstract:** Two green microalgal strains, CCCryo 469-16 and CCCryo 470-16, were isolated from samples of Antarctic microflora. Their morphology and 18S rRNA sequences indicated a phylogenetic relationship with strains of the genera *Chlorococcum* Meneghini, *Macrochloris* Korshikov, and *Pleurastrum* Chodat. The last genus is considered particularly problematic as it underwent several revisions. Moreover, its type strain, *P. insigne* SAG 30.93, was recently demonstrated to coincide genetically with the authentic strain of *Chlorococcum oleofaciens* from an 18S rRNA phylogeny. This deprived the genus *Pleurastrum* of an important type reference. Thus, the molecular markers *rbc*L, *tuf*A, and ITS were employed to identify the Antarctic isolates more precisely. Several other microalgae related to our isolates were obtained from international culture collections for comparison. The obtained results allowed the re-establishment of strain *P. insigne* SAG 30.93 as the reference strain and the definition of the molecular borders of both genera *Pleurastrum* and *Chlorococcum*. Based on our findings, several *Chlorococcum* species are now re-attributed to *Pleurastrum*, as well as *Macrochloris rubrioleum*, here re-named *Pleurastrum rubrioleum* comb. nov., to which the Antarctic isolates also belong. Finally, a relationship between *Pleurastrum* lineages and climate zones was established.

**Keywords:** 18S rRNA; *Chlorococcum*; climate zones; ITS2 secondary structure; *Macrochloris rubrioleum*; *Pleurastrum*; *rbc*L; *tuf*A

## 1. Introduction

Under the term "green microalgae" are included different photoxygenic eukaryotic microorganisms, mainly belonging to the phylum Chlorophyta, which can be found in freshwater, marine, and terrestrial environments, as well as in environments considered extreme, such as deserts and polar regions [1,2]. In particular, in cold habitats, such as alpine snowfields and poles, characterized by limiting conditions such as low temperature, nutrient scarcity, and high UV irradiation, there are several species of green microalgae, and some of them, often characterized by orange to red color for the large amount of carotenoids, are also colloquially called "snow algae", e.g., [3–7].

To face the extreme conditions of their habitats, cold-adapted microalgae have developed morphological strategies, such as the formation of robust secondary cell walls and latent overwintering stages, e.g., [5], and metabolic adaptations, such as the production of high amounts of polyunsaturated fatty acids and the accumulation of carotenoids, e.g., [4,6]. In general, most of the biochemical compounds produced by microalgae can be employed in different human industries, such as the pharmaceutical, cosmeceutical, and nutraceutical fields, e.g., [8–10], and in particular, snow microalgae have shown a great potential to produce interesting biomolecules, e.g., [4,11]. As previously highlighted, e.g., [8,9], the compound production capability of microalgae can vary among taxa, even below the species

rank, thus underlying how the correct taxonomic identification of microalgal isolates is a prerequisite also for practical purposes.

The whole biodiversity of green microalgae is still to be revealed, and new evolutionary lineages and species are continuously discovered and described, particularly in underinvestigated environments such as Antarctica, e.g., [7,12–15].

During sampling campaigns aimed at screening the Antarctic microflora biodiversity, also with the purpose of investigating its possible biochemical exploitation, two green coccoid microalgae were isolated from the Victoria Land region (Antarctica). In this work, results concerning the molecular analyses carried out to identify the two isolates are shown. Analyses were based on different molecular markers: the nuclear 18S rRNA gene, the plastid *rbc*L (encoding for the large subunit of the Rubisco enzyme) and *tuf*A (encoding for the elongation factor Tu) genes, and the nuclear internal transcribed spacer (ITS) region (a genomic portion between the 18S and 28S rRNA genes), with a particular focus on its ITS2 secondary structures. The obtained results led us to attribute the two Antarctic strains to the genus *Pleurastrum* Chodat.

The genus *Pleurastrum* was first described by Chodat in 1894 [16], and despite its simple morphology, a high phenotypic plasticity was successively reported among the different species of this taxon. In particular, solitary cells (i.e., coccoid form), more complex filamentous forms, and pseudoparenchymatous aggregates of cells, called "sarcinoid packets", are morphologies attributed to this green microalgal group [17]. The strong environmental influence on *Pleurastrum* morphology and the loss of cultured material for the type species *Pleurastrum insigne* Chodat have led to problems in the delimitation of this debated genus [18]. Thus, over time, a number of isolates were erroneously attributed to this taxon, which reached a total of ten species [19]. Later, the genus was revised, with several species transferred to other taxa, and currently, only four species are recognized under *Pleurastrum*: *P. insigne* (re-discovered, re-isolated, and lectotypified in 1990 by Sluiman and Gärtner [18]), *Pleurastrum photoheterotrophicum* Metting (even if this name is probably invalid since the word "type" was not present in the original description), *Pleurastrum sarcinoideum* Groover & Bold, and *Pleurastrum terricola* (Bristol) D.M. John [19]. According to Algaebase [19], currently, *Pleurastrum* is the only genus in the family Pleurastraceae, order Chlamydomonadales, class Chlorophyceae, phylum Chlorophyta. Skaloud and collaborators [17], instead, place *Pleurastrum* likely in the family Helicodictyaceae, order Ulotrichales, class Ulvophyceae, phylum Chlorophyta; moreover, these authors list just three species under the genus (i.e., *P. insigne*, *P. photoheterotrophicum*, and *P. sarcinoideum*) since *P. terricola* is considered a synonym of *Pleurastrum terrestre*, now transferred to the genus *Leptosira* Borzì.

In 2015, Kawasakii and collaborators [20] carried out a study on *Chlorococcum oleofaciens* Trainor & Bold and allied species belonging to the genus *Chlorococcum* Meneghini, a taxon morphologically very similar to *Pleurastrum*, particularly at a first glance. In that study [20], based on an 18S rRNA phylogeny, Kawasakii et al. showed that the authentic strain of *P. insigne* (indicated by Sluiman and Gärtner [18] and preserved in the Culture Collection of Algae at the University of Goettingen (SAG), Germany, under the code SAG 30.93) was genetically identical to *C. oleofaciens* SAG 213-11, the reference strain for the species *C. oleofaciens*. Kawasakii and collaborators [20], therefore, set cultures of *P. insigne* SAG 30.93, trying to reproduce the same conditions used by Sluiman and Gärtner [18], in order to verify if strain SAG 30.93 morphologically fitted into the *P. insigne* description. Since Kawasakii and collaborators [20] did not observe the filamentous form classically reported for *P. insigne*, the authors concluded that strain SAG 30.93 had been overgrown by *C. oleofaciens*. This deprived the genus *Pleurastrum* of an important reference, resulting in a tough blow for the systematics of this taxon. Nevertheless, Kawasakii et al. [20] did not include the authentic strain of the type species of *Chlorococcum*, *Chlorococcum infusionum* (Schrank) Meneghini, (i.e., strain SAG 10.86) in their analyses; this would have indicated the true *Chlorococcum* clade.

Besides the taxonomic identification of the two Antarctic isolates, given the several green microalgal strains obtained from international culture collections for comparison during this work, the achieved results allowed us to re-establish the existence of the *P. insigne* authentic strain and to enlarge the taxonomic borders of *Pleurastrum*, including under this genus several species previously attributed to *Chlorococcum*.

## 2. Materials and Methods

### 2.1. Sampling and Culture Set up of Antarctic Microalgal Strains

During two scientific expeditions (1993–1994 and 2003–2004) in Antarctica, several microalgal samples were collected. In particular, a coccoid chlorophycean strain, tagged as "04/16B", was sampled in a water pond near the OASI telescope hill, at the Mario Zucchelli Station, Terra Nova Bay, Victoria Land (−74.695, 164.123), and another coccoid chlorophycean strain, named "A08", was isolated from a greenish snow sample collected at Edmonson Point, Wood Bay, Victoria Land (−74.333, 165.133) (Figure 1). Cultures of strains 04/16B and A08 were set up in both liquid media Czurda [21] and BG11 [22] and kept at both 4 °C and 16 °C, under a continuous light intensity of 10–12 µmol photons m$^{-2}$ s$^{-1}$. The isolated Antarctic strains were deposited at the Culture Collection of Cryophilic Algae (CCCryo), Fraunhofer, Germany; strain 04/16B was attributed the code CCCryo 469–16 and strain A08 the code CCCryo 470–16.

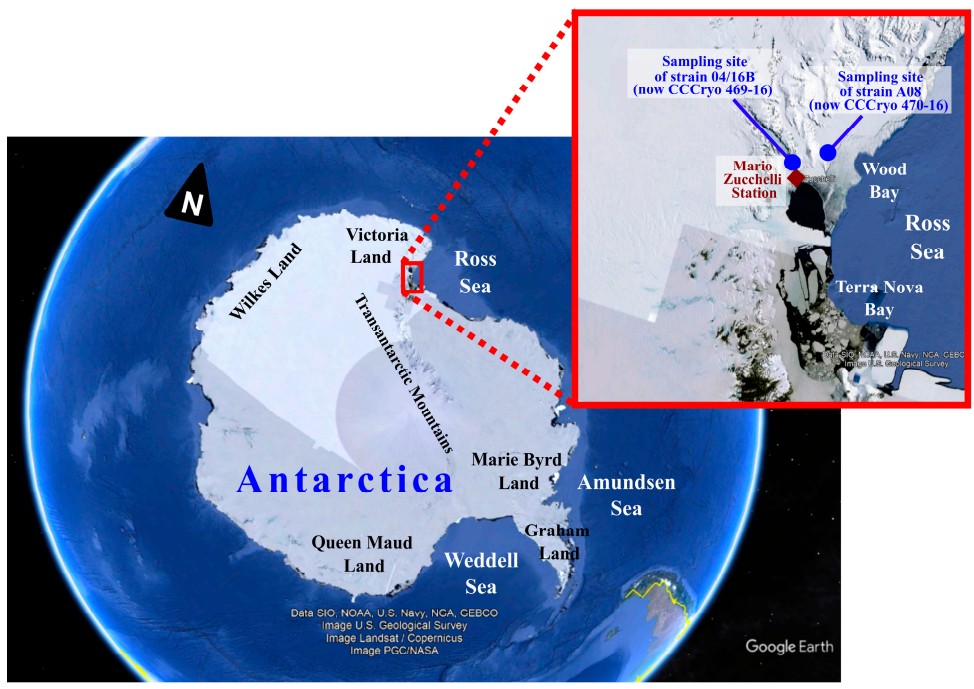

**Figure 1.** Map of Antarctica showing the sampling sites of strains CCCryo 469-16 and CCCryo 470-16.

### 2.2. Taxon Selection, Culture Conditions, and Light Microscope Observations

In addition to the Antarctic strains CCCryo 469-16 and CCCryo 470-16, 18 microalgal strains belonging to the phylum Chlorophyta were selected and obtained from international culture collections (i.e., the Culture Collection of Algae at the University of Goettingen (SAG), Germany; the Culture Collection of Cryophilic Algae (CCCryo) at the Fraunhofer Institute for Cell Therapy and Immunology IZI, Potsdam, Germany; and the Culture Collection of Algae at the University of Texas (UTEX), Austin, TX, USA) to be used for comparison (Table 1); most of them were authentic strains, and two organisms (CCCryo 194-04 and CCCryo 205-05) were Arctic strains attributed to the genus *Chlorococcum* and deposited at the CCCryo culture collection. The cold-adapted strains were kept in culture with BG11 liquid medium, at 4 °C, and under a continuous light inten-

sity of 10–12 µmol photons $m^{-2} s^{-1}$. The other strains were maintained following the recommendations of the respective culture collections.

**Table 1.** Focal strains, listed in alphabetical order, for which at least one sequence was produced in this study. For each strain, the original designation with which it was found in the corresponding culture collection is indicated, as well as the corresponding INSD accession numbers of the sequences obtained for the considered molecular markers (18S rRNA, *rbc*L, *tuf*A, and ITS region). The strains obtained from international culture collections are available from the following collections: SAG = Culture Collection of Algae at the University of Goettingen, Germany; CCCryo = Culture Collection of Cryophilic Algae at the Fraunhofer Institute for Cell Therapy and Immunology IZI, Potsdam, Germany; UTEX = Culture Collection of Algae at the University of Texas, Austin, USA. With the exception of the strains sampled in this study and of strains CCCryo 194-04 and CCCryo 205-05, all the listed organisms are authentic strains. The sequences obtained in this study are in boldface font; a dash denotes that no data are available. The two new Antarctic isolates are indicated in boldface font and with an asterisk.

| Taxon | Strain Identifier | 18S rRNA | *rbc*L | *tuf*A | ITS |
|---|---|---|---|---|---|
| *Chlorococcum citriforme* | SAG 62.80 | KM020100 | - | **LT594531** | **LT594562** |
| *Chlorococcum costatozygotum* | SAG 20.95 | - | - | **LT594535** | **LT594566** |
| *Chlorococcum diplobionticum* | SAG 32.95 | - | - | **LT594536** | **LT594567** |
| *Chlorococcum echinozygotum* | SAG 213-5 | KM020131 | EF113430 | **LT594526** | **LT594557** |
| *Chlorococcum hypnosporum* | SAG 213-6 | JN904003 | **LT594545** | **LT594527** | **LT594558** |
| *Chlorococcum infusionum* | SAG 10.86 | KM020174 | **LT594548** | **LT594534** | **LT594565** |
| *Chlorococcum isabeliense* | SAG 65.80 | KM020106 | - | **LT594532** | **LT594563** |
| *Chlorococcum minutum* | SAG 213-7 | KM020099 | **LT594546** | **LT594528** | **LT594559** |
| *Chlorococcum oleofaciens* | SAG 213-11 | AB983608 | - | **LT594530** | **LT594561** |
| *Chlorococcum sphacosum* | SAG 66.80 | KM020102 | **LT594547** | **LT594533** | **LT594564** |
| *Chlorococcum tatrense* | UTEX 2227 | - | **LT594550** | **LT594538** | **LT594569** |
| *Chlorococcum vacuolatum* | SAG 213-8 | KM020107 | - | **LT594529** | **LT594560** |
| *Chlorococcum* sp. | CCCryo 194-04 | - | **LT594551** | **LT594539** | HQ404881 |
| *Chlorococcum* sp. | CCCryo 205-05 | - | - | **LT594540** | HQ404882 |
| *Fasciculochloris boldii* | SAG 27.95 | - | **LT594552** | **LT594541** | **LT594570** |
| *Macrochloris rubrioleum* | CCCryo 340b-08 | GU117573 | - | **LT989896** | AB983643 |
| *Pleurastrum insigne* | SAG 30.93 | AB983614 | EF113464 | **LT594537** | **LT594568** |
| *Pleurastrum* **sp. 04/16B *** | **CCCryo 469-16** | **LT594553** | **LT594543** | **LT594524** | **LT594555** |
| *Pleurastrum* **sp. A08 *** | **CCCryo 470-16** | **LT594554** | **LT594544** | **LT594525** | **LT594556** |
| *Tetracystis aeria* | SAG 89.80 | JN903990 | EF113476 | LT594542 | - |

Light microscopy (LM) observations were carried out both on the sampled strains and on the strains obtained from culture collections, using a Leica 5000 microscope (Wetzlar, Germany) equipped with a digital camera.

### 2.3. Electron Microscope Observations

Scanning electron microscope (SEM) and transmission electron microscope (TEM) observations were both carried out on most of the strains investigated in this work, including the two new Antarctic isolates and strains obtained from international culture collections.

For both the SEM and TEM analyses, cells were fixed in 6% (*v/v*) glutaraldehyde in 0.1 M sodium cacodylate buffer (pH 6.9) at room temperature for 2 h.

For SEM observations, the fixed cells were dehydrated with a graded ethanol series from 30% to absolute ethanol before being critical point dried with $CO_2$ using a Polaron CPD7501 (Quorum Technologies, Lewes, UK). The dried cells were then coated with gold by an Edwards S 150B Sputter Coater and observed by a scanning electron microscope Cambridge Stereoscan 260 (Cambridge, UK).

For TEM analyses, the fixed cells were washed in 0.1 M sodium cacodylate buffer (pH 6.9) twice and post-fixed in 1% (*w/v*) osmium tetroxide in 0.1 M sodium cacodylate buffer (pH 6.9) at room temperature for 2 h. The cells were dehydrated through a graded

series of ethanol followed by propylene oxide. The dehydrated cells were then embedded in araldite. Ultra-thin sections, cut with an ultramicrotome Reichert Ultracut S Leica (Vienna, Austria), were collected on copper grids, stained with uranyl acetate and lead citrate, and observed using a transmission electron microscope FEI Tecnai G2 (Hillsboro, OR, USA).

### 2.4. DNA Extraction, Amplification, and Sequencing

Culture aliquots of the considered algal strains were ground in a mortar with liquid nitrogen and genomic DNA was extracted using the Genomic DNA purification kit (Thermo Fisher Scientific, Waltham, MA, USA) according to the manufacturer's instructions.

Standard PCR protocols were carried out in 50 µL aliquots with Taq DNA polymerase (Thermo Fisher Scientific, Waltham, MA, USA), according to the manufacturer's recommendations. A portion of the 18S rRNA gene was amplified as five overlapping fragments using the primer pairs and thermocycling conditions reported in Andreoli et al. [23]. The primer pair ScenRub_F1 and ScenRub_R1 [13] was used to amplify an *rbc*L gene fragment, the primer pair tufAF and tufAR [24] was employed for the *tuf*A gene fragment, and the primer pair ITS1 [25] and ITS4 [26] was used for the ITS region; the ITS region fragment considered in the analyses included the 5.8S rRNA gene (complete sequence), the ITS2, and the first portion of the 28S rRNA gene. Thermocycling conditions to amplify the *rbc*L, *tuf*A, and ITS region *loci* followed Sciuto et al. [13].

DNA sequencing was performed at the BMR Genomics Sequencing Service (University of Padova), with the same primer pairs used in the amplification reactions. For the *rbc*L gene, full-length sequences were obtained using two additional internal sequencing primers: ScenIntr_F1 and ScenIntr_R1 [13].

The final consensus sequences were assembled using the SeqMan II program from the Lasergene software package (DNAStar©, Madison, WI, USA). The obtained sequences were deposited in the INSDC repositories, through the ENA platform, with the accession numbers reported in Table 1.

### 2.5. Molecular and Phylogenetic Analyses

The obtained sequences were compared with the sequences available in the INSDC archives with the BLAST tool [27], and different datasets were constructed for the 18S rRNA gene, the *rbc*L gene, the *tuf*A gene, and the ITS region. Each dataset included the sequences obtained in this work plus other available sequences from the INSDC repositories, choosing the most similar according to the BLAST search, and other sequences with adequate length of different genera belonging to the class Chlorophyceae (according to Algaebase [19]). Sequences obtained from representative genera of the class Trebouxiophyceae (according to Algaebase [19]) were added to root the phylogenetic reconstructions. More in detail, the 18S rRNA dataset included 35 sequences, 31 of which were from Chlorophyceae and four from Trebouxiophyceae taxa; the *rbc*L dataset was composed of 36 sequences, of which 32 were from Chlorophyceae and four from Trebouxiophyceae members; the *tuf*A dataset included 30 sequences, of which 27 were from Chlorophyceae and three from Trebouxiophyceae taxa; and the ITS region dataset was composed of 36 sequences, of which 32 were from Chlorophyceae and four from Trebouxiophyceae members.

For each dataset, a multi-alignment of sequences was generated with MUSCLE [28]. The 18S rRNA dataset had 321 parsimony informative sites in 1701 aligned positions, the *rbc*L dataset presented 193 parsimony informative sites in 604 aligned positions, the *tuf*A dataset had 243 parsimony informative sites of 709 aligned positions, and the ITS region dataset had 292 parsimony informative sites of 640 aligned positions.

Phylogenetic analyses based on the neighbor joining (NJ) and maximum parsimony (MP) methods were performed using the program MEGA version XI [29], maximum likelihood (ML) analyses were carried out with PHYML version 3.065 [30], and Bayesian inference (BI) analyses were performed with MrBayes version 3.1.2 [31]. Nonparametric bootstrap (BT) re-sampling [32] was used to test the robustness of the NJ, MP, and ML tree topologies (1000 BT replicates).

For the ML and BI analyses, the models that best fit our data were found using jModelTest version 0.1.1 [33–35], under the BIC criterion [36]. The model that best fit the 18S rRNA data was TIM1 + I + G, and the following parameters were implemented: nucleotide frequencies as freqA = 0.2533, freqC = 0.2055, freqG = 0.2721, and freqT = 0.2691; substitution rate matrix with A-C substitutions = 1.0000, A-G = 3.0154, A-T = 1.7343, C-G = 1.7343, C-T = 7.9307, and G-T = 1.0000; proportion of sites assumed to be invariable = 0.4710; and gamma shape = 0.4440. For the *rbc*L dataset, jModelTest indicated GTR + G as the best model, and the following parameters were implemented: nucleotide frequencies as freqA = 0.2596, freqC = 0.1672, freqG = 0.2131, and freqT = 0.3602; substitution rate matrix with A-C substitutions = 0.4715, A-G = 1.2019, A-T = 3.0585, C-G = 0.6121, C-T = 4.5689, and G-T = 1.0000; proportion of sites assumed to be invariable = 0; and gamma shape = 0.2300. For the *tuf* A dataset, the model that best fit the data was GTR + I + G, and the following parameters were implemented: nucleotide frequencies as freqA = 0.3586, freqC = 0.1171, freqG = 0.1945, and freqT = 0.3298; substitution rate matrix with A-C substitutions = 2.6292, A-G = 4.5896, A-T = 6.5369, C-G = 3.4215, C-T = 16.1411, and G-T = 1.0000; proportion of sites assumed to be invariable = 0.3970; and gamma shape = 0.9840. For the ITS region dataset, the model that best fit the data was SYM + G, and the following parameters were implemented: substitution rate matrix with A-C substitutions = 0.8819, A-G = 2.0353, A-T = 2.1083, C-G = 0.6897, C-T = 4.2819, and G-T = 1.0000; proportion of sites assumed to be invariable = 0; and gamma shape = 0.3870.

BI analyses included two independent concurrent MCMC runs, each composed of four chains, three heated and one cold. Each MCMC ran for $1 \times 10^6$ generations (18S rRNA data), $15 \times 10^6$ generations (*rbc*L data), $1.5 \times 10^6$ generations (*tuf* A data), and $2 \times 10^6$ generations (ITS region data), sampling trees every 100 generations. We considered the sampling of the posterior distribution to be adequate if the average standard deviation of split frequencies was ≤0.01. The first 2500 trees for the 18S rRNA gene, 37,500 trees for the *rbc*L gene, 3750 trees for the *tuf* A gene, and 5000 trees for the ITS region were discarded as burn-in, as determined by stationarity of lnL assessed using Tracer version 1.5 [37]. Consensus topologies and posterior probabilities (PP) values were then calculated from the remaining trees. The nexus files for the BI analyses were generated through Mesquite version 2.71 [38].

### 2.6. ITS2 Secondary Structure Predictions and Analyses

For the strains placed inside the *Pleurastrum* clade in the performed phylogenetic reconstructions, the ITS2 portion of the ITS region was analyzed more in detail. For the newly obtained ITS region sequences and for the ITS sequences retrieved from the INSDC archives, ITS2 boundaries were detected using the tool "Annotate" implemented in the ITS2 Database [39–42]. In some cases, the ITS2 Database could not find ITS2 boundaries; thus, the start and end points of the ITS2 were determined by comparison with other related ITS2 sequences, looking for the flanking 5.8S and 28S rRNA genes.

For each ITS2 sequence, the corresponding secondary structure was obtained with the tool "Predict" of the ITS2 Database. In most of the cases, secondary structures were obtained by direct fold and, in few cases, by homology modeling.

An ITS2 dataset was created including 18 sequences of strains attributed to the genus *Pleurastrum* and their corresponding secondary structures, and it was used to obtain an ITS2 sequence + secondary structure multi-alignment with both the programs 4SALE version 1.7 [43,44] and MARNA [45]. The two resulting sequence + secondary structure multi-alignments were carefully inspected, compared, and manually adjusted to obtain a final consensus sequence + secondary structure multi-alignment. From this final multi-alignment, the ITS2 consensus secondary structure of the genus *Pleurastrum* was inferred and successively drawn with the program VARNA version 3.9 [46]. Compensatory base changes (CBCs) among the different *Pleurastrum* strains were detected on the ITS2 sequence + secondary structure final multi-alignment using both the programs 4SALE and CBCAnalyzer [47]; the latter allowed also the identification of hemi-compensatory base

changes (hCBCs). A phylogenetic reconstruction was also carried out on the ITS2 sequence + secondary structure final multi-alignment with the program ProfDistS Qt_version 0.9.9 [48]. The phylogenetic tree was obtained performing the NJ algorithm [49], using the GTR model and the rate matrix "Q_ITS2.txt" for distance correction, with 1000 bootstrap replicates. Moreover, the multi-alignment of the only ITS2 sequences was exported and used for MP and ML analyses with MEGA. For the ML analyses the K2 + I model was applied, as selected by the "Find best DNA Models" tool implemented in MEGA, under the BIC criterion.

Following the method proposed by Darienko et al. [50], 87 barcode sites were detected on the *Pleurastrum* ITS2 consensus secondary structure, and each barcode site was codified into a number representing the ITS2 sequence + secondary structure information (A-U = 1, U-A = 2, G-C = 3, C-G = 4, G•U = 5, U•G = 6, mismatch = 7, deletion or unpaired or single bases = 8). The numeric barcodes generated in this way were manually aligned, and the obtained numeric barcode alignment was transformed in nexus format with Mesquite and used for an MP analysis with PAUP version 4.0a [51], testing the robustness of the obtained topologies by BT re-sampling (1000 replicates).

The ITS2 multi-alignment of only sequences was also used to apply the following species delimitation methods: the automatic barcode gap discovery (ABGD) method [52], the assemble species by automatic partitioning (ASAP) method [53], the Poisson tree processes (PTP) method [54], and the generalized mixed Yule coalescent (GMYC) method [55]. AGBD was performed using the web interface https://bioinfo.mnhn.fr/abi/public/abgd/abgdweb.html (accessed on 30 November 2022); Kimura (K80) with TS/TV = 2.0 was selected as the distance model, and the relative gap width (X) was set at 0.5. ASAP was carried out through the https://bioinfo.mnhn.fr/abi/public/asap/asapweb.html online platform (accessed on 2 December 2022); again Kimura (K80) with TS/TV = 2.0 was selected as the distance model. PTP was run on the web server http://species.h-its.org/ptp (accessed on 5 December 2022), leaving the default values, except for the generation number that was set to 500,000, and using the ML phylogenetic tree (without BT statistics) previously obtained with MEGA as the input. GMYC was carried out on the http://species.h-its.org/gmyc web interface (accessed on 6 December 2022), using an ultrametric tree as the input and selecting the "single threshold" method. For the GMYC test, the ultrametric tree was obtained with BEAST version 2.6.3.0 [56], using the GTR + I model; the "log normal relaxed molecular clock" was selected as the clock model, a "coalescence tree with constant population" was chosen as the tree model/prior, and the MCMC was run for $1 \times 10^6$ generations, sampling trees every 100 generations and finally discarding the first 2500 trees as burn-in. TreeAnnotator version 1.8.0 [57] and FigTree version 1.4.4 (available at http://tree.bio.ed.ac.uk/software/figtree/; accessed on 7 December 2022) were used to obtain the final ultrametric tree.

Finally, the ITS2 only sequence multi-alignment was used to study and visualize the *Pleurastrum* haplotypes with the program PopART version 1.7 [58], selecting the minimum spanning network as the network inference method [59].

The final publishable images were created with CorelDRAW Graphics Suite X4, Inkscape version 0.92 (https://www.inkscape.org; accessed on 4 January 2023), and GIMP version 2.8.22 (https://www.gimp.org; accessed on 4 January 2023).

## 3. Results

### 3.1. Morphological Observations

The two Antarctic isolates CCCryo 469-16 and CCCryo 470-16, kept at the same above reported culture conditions, had similar morphologies if observed using the light microscope (Figure 2). Cells were mainly spherical, with a few more ellipsoidal cells, and generally solitary, even if groups of three (triads) and four cells (tetrads) could be observed (Figure 2A). In the cells, the chloroplast occupied almost all the cell cytoplasm, and in many cells, a pyrenoid was visible (Figure 2A). Vegetative cells ranged in size between 5 and 15 µm in diameter (Figure 2A,B), while in the sporangium phase, cells reached bigger sizes,

being about 10 to 20 μm in diameter (Figure 2C) and containing several daughter cells when mature (Figure 2D). Inside sporangia, the daughter cells could have different and irregular morphologies (Figure 2C). Ellipsoid zoospores, about 2.5 μm wide and 5 μm long, were observed (Figure 2E). Just after their isolation and during the first days in culture, the two Antarctic isolates showed also sarcinoid packets and occasional short filamentous-like phases (data not shown), which were never observed later.

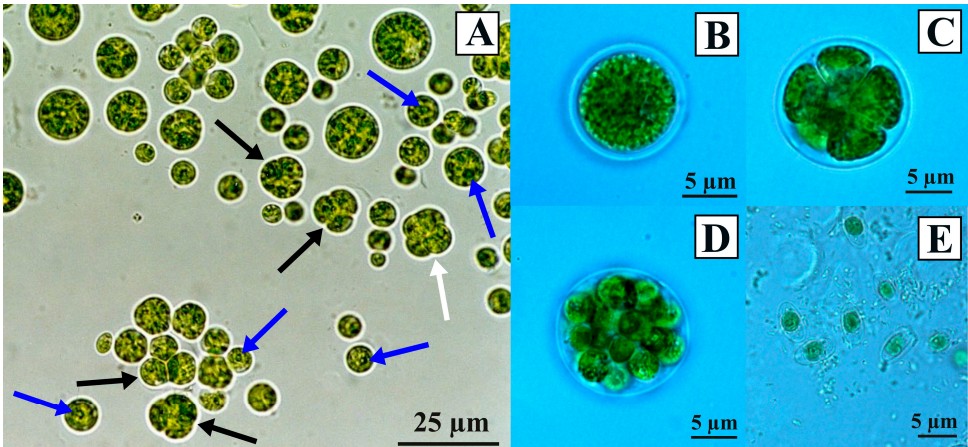

**Figure 2.** Light microscope images of the Antarctic strain CCCryo 470-16. (**A**) General view showing the different cell sizes of the microalga in the same culture condition. The black arrows indicate cell triads, the white arrow indicates a cell tetrad, and the blue arrows indicate a visible pyrenoid inside the chloroplasts. (**B**) Vegetative cells. (**C**) A sporangium in its initial stage. (**D**) A mature sporangium. (**E**) Zoospores.

The features observed using the light microscope for the two Antarctic isolates were present also in the strains obtained from international culture collections (Figure S1), which had generally spherical cells, with a few strains showing more ellipsoidal or anyway elongated cells (e.g., *Chlorococcum citriforme* SAG 62.80 in Figure S1A and *Chlorococcum isabeliense* SAG 65.80 in Figure S1G). Also in the case of the focal strains obtained from international culture collections, sporangia were composed of daughter cells with various sizes and forms (e.g., *Chlorococcum infusionum* SAG 10.86 in Figure S1F and *Chlorococcum vacuolatum* SAG 213-8 in Figure S1M), and cell aggregates, kept together by extracellular polysaccharides, were observed (e.g., *Chlorococcum minutum* SAG 213-7 in Figure S1H and *Chlorococcum oleofaciens* SAG 213-11 in Figure S1I).

Observations using the scanning electron microscope (Figure 3) confirmed the spherical and sometimes ellipsoid cell shape in all the focal strains and the formation of cell aggregates kept together from extracellular polysaccharides (e.g., *Chlorococcum oleofaciens* SAG 213-11 in Figure 3F and *Chlorococcum diplobionticum* SAG 32.95 in Figure 3M).

*3.2. Ultrastructural Observations on the Antarctic Isolates*

Using the transmission electron microscope (Figure 4), the two Antarctic isolates CCCryo 469-16 and CCCryo 470-16, kept at the same above reported culture conditions, had similar ultrastructures. Vegetative cells were spherical and had a cell diameter of about 5 μm, a thick cell wall, and a parietal chloroplast with a single pyrenoid inside (Figure 4A). The starch plate surrounding the pyrenoid was always continuous and not crossed by thylakoids (Figure 4A–D). Dyads of cells, originated from the division of a mother cell at the cell equator (Figure 4B), and sporangia formed by cell tetrads (Figure 4C) were present. The size of mature sporangia was higher than that of vegetative cells, reaching 12–15 μm in diameter (Figure 4D). Mature sporangia included often more than four new forming cells, as in the case of Figure 4D, where a mature sporangium made up of seven cells, each surrounded by a new cell wall and with different sizes and shapes, was visible.

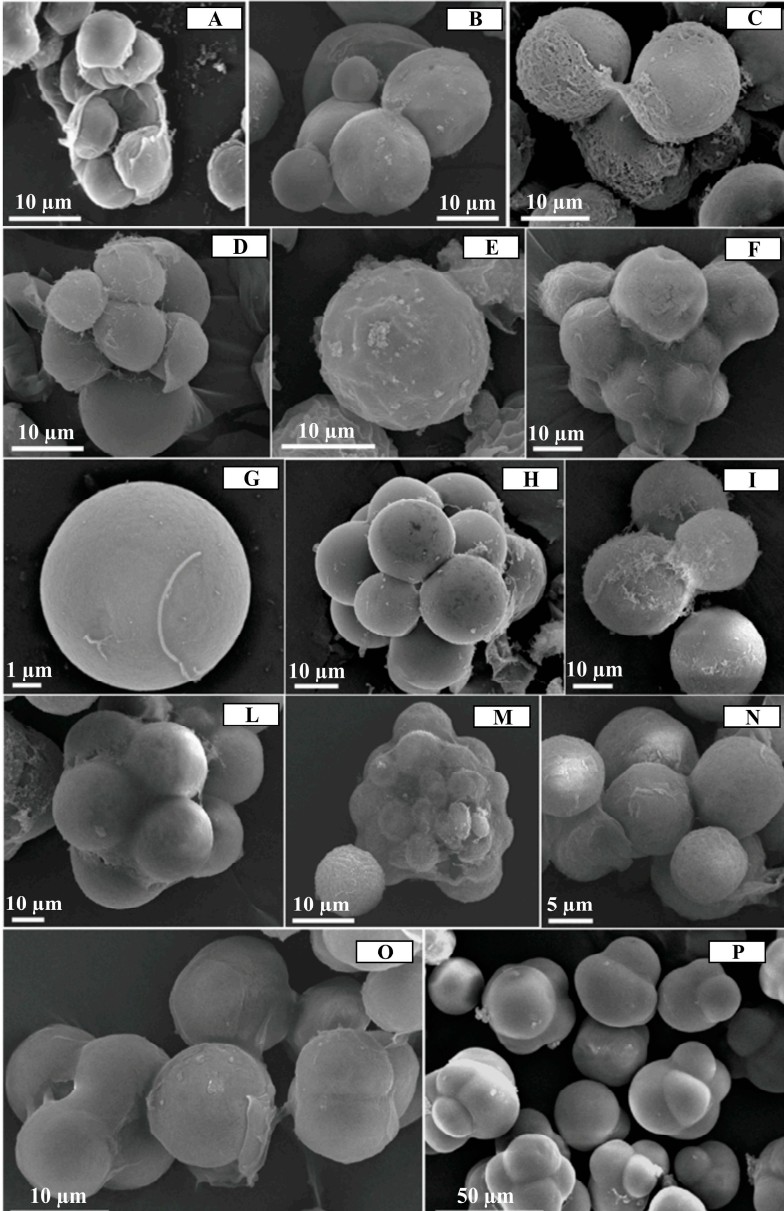

**Figure 3.** Overview, using the scanning electron microscope, of several microalgal strains considered in this study. (**A**) The Antarctic strain CCCryo 469-16, (**B**) *Chlorococcum echinozygotum* SAG 213-5, (**C**) *Chlorococcum hypnosporum* SAG 213-6, (**D**) *Chlorococcum minutum* SAG 213-7, (**E**) *Chlorococcum vacuolatum* 213-8, (**F**) *Chlorococcum oleofaciens* SAG 213-11, (**G**) *Chlorococcum isabeliense* SAG 65.80, (**H**) *Chlorococcum sphacosum* SAG 66.80, (**I**) *Chlorococcum infusionum* SAG 10.86, (**L**) *Chlorococcum costatozygotum* SAG 20.95, (**M**) *Chlorococcum diplobionticum* SAG 32.95, (**N**) *Chlorococcum* sp. CCCryo 194-04, (**O**) *Chlorococcum* sp. CCCryo 205-05, and (**P**) *Tetracystis aeria* SAG 89.80.

### 3.3. Molecular and Phylogenetic Results

In the phylogenetic reconstruction based on the 18S rRNA gene fragment (Figure 5), strains CCCryo 469-16 and CCCryo 470-16 and the other green microalgal strains focused on in this study were scattered among two well-supported clades, which were named clade A (support values: 100/99/100/1.00) and clade B (support values: 100/99/100/1.00).

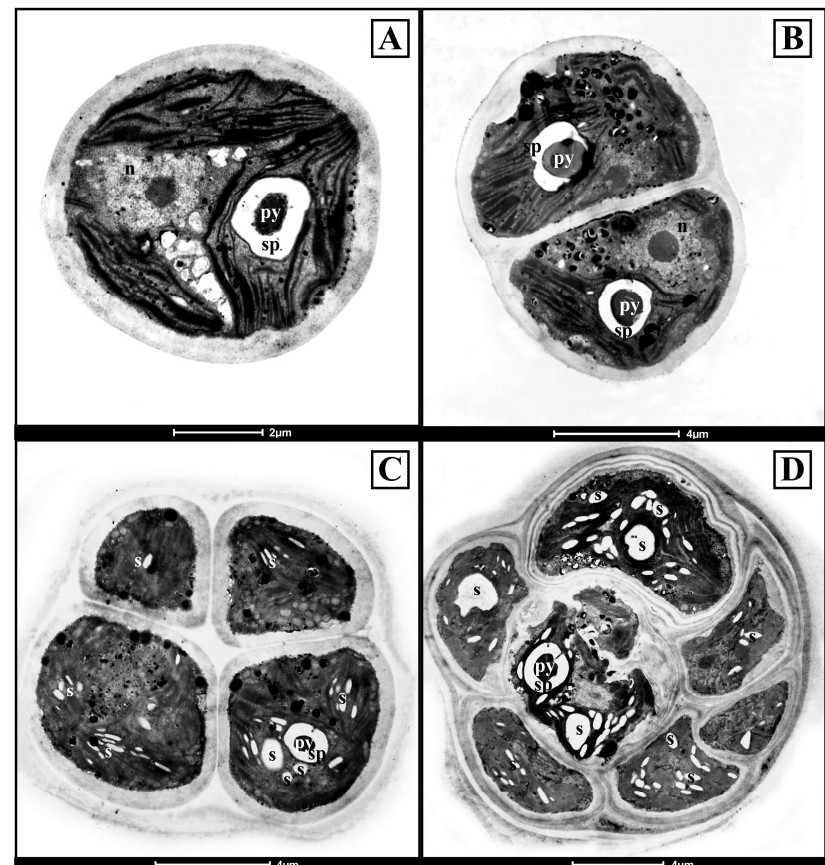

**Figure 4.** Scanning electron microscope micrographs of the Antarctic strain CCCryo 469-16. (**A**) A single coccoid cell; (**B**) a dyad, formed just after cell division; (**C**) a sporangium forming a tetrad; and (**D**) a sporangium containing seven cells. n = nucleus; py = pyrenoid; sp = starch plate; s = starch.

Clade A (Figure 5) included most of the focal strains: strains CCCryo 469-16 and CCCryo 470-16, as well as two other strains isolated from Antarctica (*Chroococcales* sp. strain II4 and *Chroococcales* sp. strain VI8), *Pleurastrum insigne* SAG 30.93 (authentic strain of the type species of the genus *Pleurastrum*), *Macrochloris rubrioleum* CCCryo 340b-08, *Macrochloris rubrioleum* CCCryo 006-99, *Chlorococcum sphacosum* SAG 66.80, *Chlorococcum oleofaciens* SAG 213-11, *Chlorococcum citriforme* SAG 62.80, *Chlorococcum microstigmatum* UTEX 1777, *Chlorococcum vacuolatum* SAG 213-8, *Chlorococcum isabeliense* SAG 65.80, *Chlorococcum rugosum* UTEX 1785, *Chlorococcum minutum* SAG 213-7, *Chlorococcum aquaticum* UTEX 2222, and *Chlorococcum* sp. CCAP 11/52.

Inside clade A (Figure 5), six lineages could be detected: (1) *P. insigne* SAG 30.93, *C. oleofaciens* SAG 213-11, *C. citriforme* SAG 62.80, and *C. sphacosum* SAG 66.80; (2) *C. microstigmatum* UTEX 1777 and *Chlorococcum* sp. CCAP 11/52; (3) *M. rubrioleum* CCCryo 340b-08, *M. rubrioleum* CCCryo 006-99, and the Antarctic strains CCCryo 469-16, 470-16, II4, and VI8; (4) *C. isabeliense* SAG 65.80 and *C. rugosum* UTEX 1785; (5) *C. minutum* SAG 213-7 and *C. aquaticum* UTEX 2222; and (6) *C. vacuolatum* SAG 213-8.

Clade B (Figure 5) included *Chlorococcum infusionum* SAG 10.86 (the authentic strain of the type species of the genus *Chlorococcum*), *Chlorococcum echinozygotum* SAG 213-5, *Chlorococcum hypnosporum* SAG 213-6, and *Tetracystis aeria* SAG 89.80, with *C. infusionum* SAG 10.86 and *C. echinozygotum* SAG 213-5 clustering together.

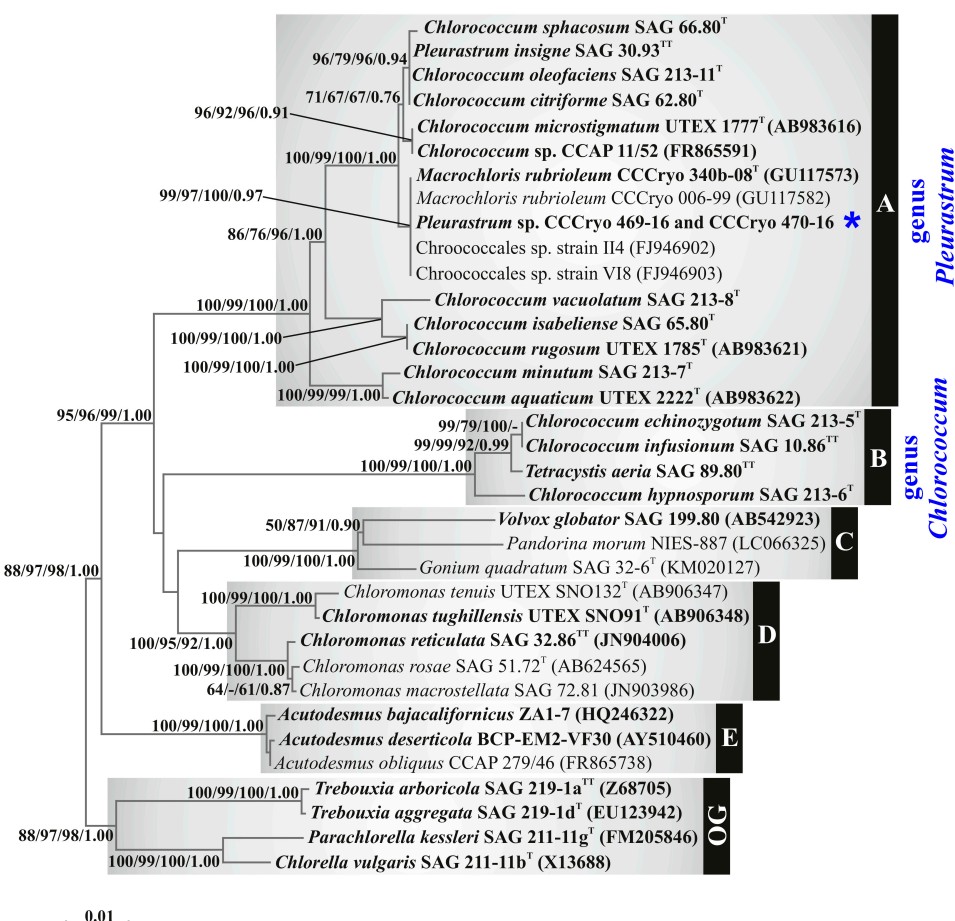

**Figure 5.** Phylogenetic reconstruction based on the 18S rRNA gene fragment. Numbers associated with nodes indicate support values for NJ, MP, ML, and BI analyses. Only bootstrap supports ≥50% and posterior probabilities ≥0.70 are reported. Values for nodes that obtained support in only one of the performed phylogenetic analyses were omitted. The horizontal bar represents the expected number of nucleotide substitutions per site. Gray boxes represent the detected clades, tagged on the right with a capital letter from A to E; "OG" stands for "outgroup". Strains common to all the phylogenetic reconstructions are highlighted in boldface font. The superscript letter "T" near a strain name means that it is the authentic strain of the corresponding species; a double superscript letter "T" means that it is the authentic strain of a type species. The two new Antarctic isolates are indicated with a blue asterisk. The genera *Pleurastrum* and *Chlorococcum*, as circumscribed in this study, are also indicated.

In the phylogenetic reconstruction based on the *rbc*L gene fragment (Figure 6), the focal strains (i.e., the two Antarctic isolates and the strains obtained from international culture collections for comparison) were again distributed into two clades, which were tagged as clade A (support values: 99/72/85/0.98) and clade E (support values: 100/99/100/1.00).

Clade A (Figure 6) was made up of *P. insigne* SAG 30.93, *Chlorococcum tatrense* UTEX 2227, *C. sphacosum* SAG 66.80, *C. minutum* SAG 213-7, the Antarctic CCCryo 469-16 and CCCryo 470-16, the Arctic strain *Chlorococcum* sp. CCCryo 194-04, two strains (UTEX 972 and LUCC 005) attributed to the species *Chlorococcum ellipsoideum* Deason & Bold, *Chlorococcum* sp. LU9, and *Chlorococcum* sp. LUCC 006.

Inside clade A (Figure 6), strains CCCryo 469-16, CCCryo 470-16, and CCCryo 194-04 formed a well-supported monophyletic clade (support values: 100/100/100/1.00) that was a sister taxon to the high-supported monophyletic clade formed by *P. insigne* SAG 30.93, *C. tatrense* UTEX 2227, and *C. sphacosum* SAG 66.80 (support values: 100/100/100/1.00).

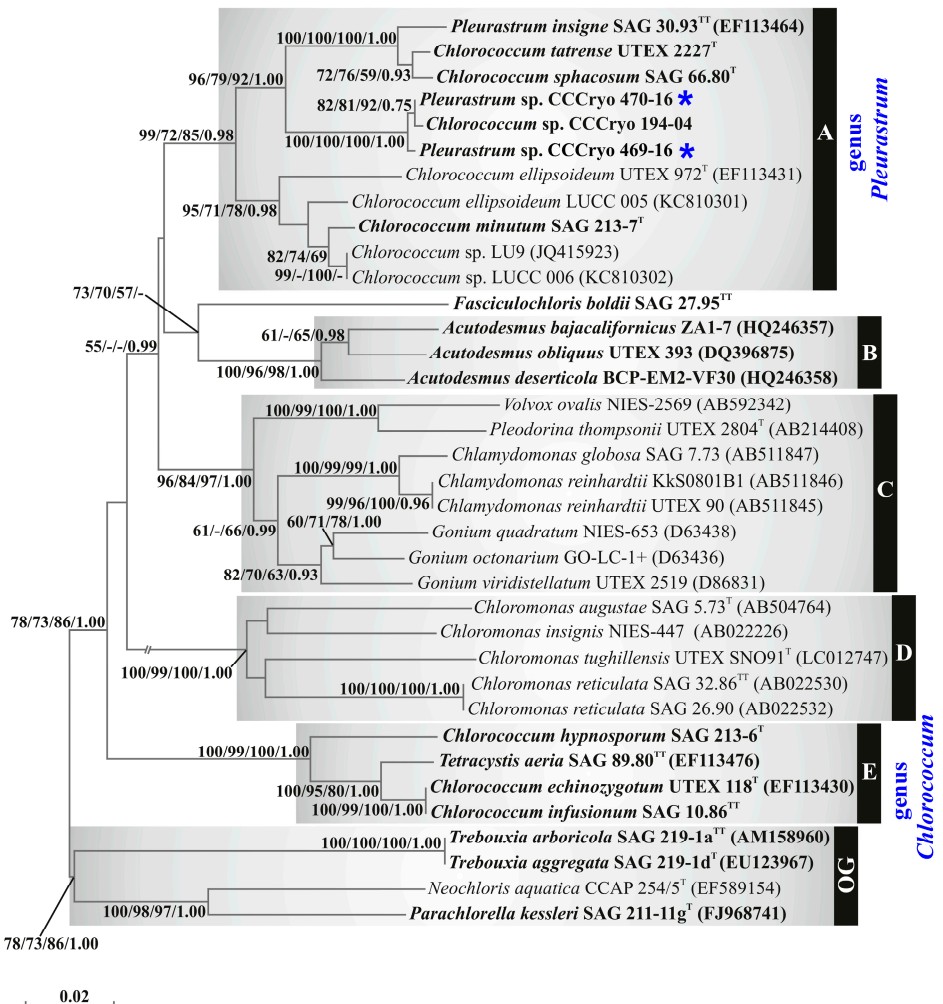

**Figure 6.** Phylogenetic reconstruction based on the *rbc*L gene fragment. Numbers associated with nodes indicate support values for NJ, MP, ML, and BI analyses, respectively. Only bootstrap supports ≥50% and posterior probabilities ≥0.70 are reported. Values for nodes that obtained support in only one of the performed phylogenetic analyses were omitted. The horizontal bar represents the expected number of nucleotide substitutions per site. Gray boxes represent the detected clades, tagged on the right with a capital letter from A to E; "OG" stands for "outgroup". Strains common to all the phylogenetic reconstructions are highlighted in boldface font. The superscript letter "T" near a strain name means that it is the authentic strain of the corresponding species; a double superscript letter "T" means that it is the authentic strain of a type species. The two new Antarctic isolates are indicated with a blue asterisk. The genera *Pleurastrum* and *Chlorococcum*, as circumscribed in this study, are also indicated.

Clade E (Figure 6) had a high statistical support (support values: 100/99/100/1.00) and was composed of *C. infusionum* SAG 10.86, *C. echinozygotum* SAG 213-5, *C. hypnosporum* SAG 213-6, and *T. aeria* SAG 89.80, with again *C. infusionum* SAG 10.86 and *C. echinozygotum* SAG 213-5 forming a unique lineage (support values: 100/99/100/1.00).

The phylogenetic reconstruction based on the *tuf*A gene (Figure 7) included the sequences of the two Antarctic isolates and of all the strains obtained from international culture collections for comparison. Again, the focal microalgae (i.e., the two Antarctic isolates and the strains obtained from international culture collections for comparison) were divided into two highly supported clades, named A (support values: 99/83/81/0.99) and C (support values: 100/100/100/1.00).

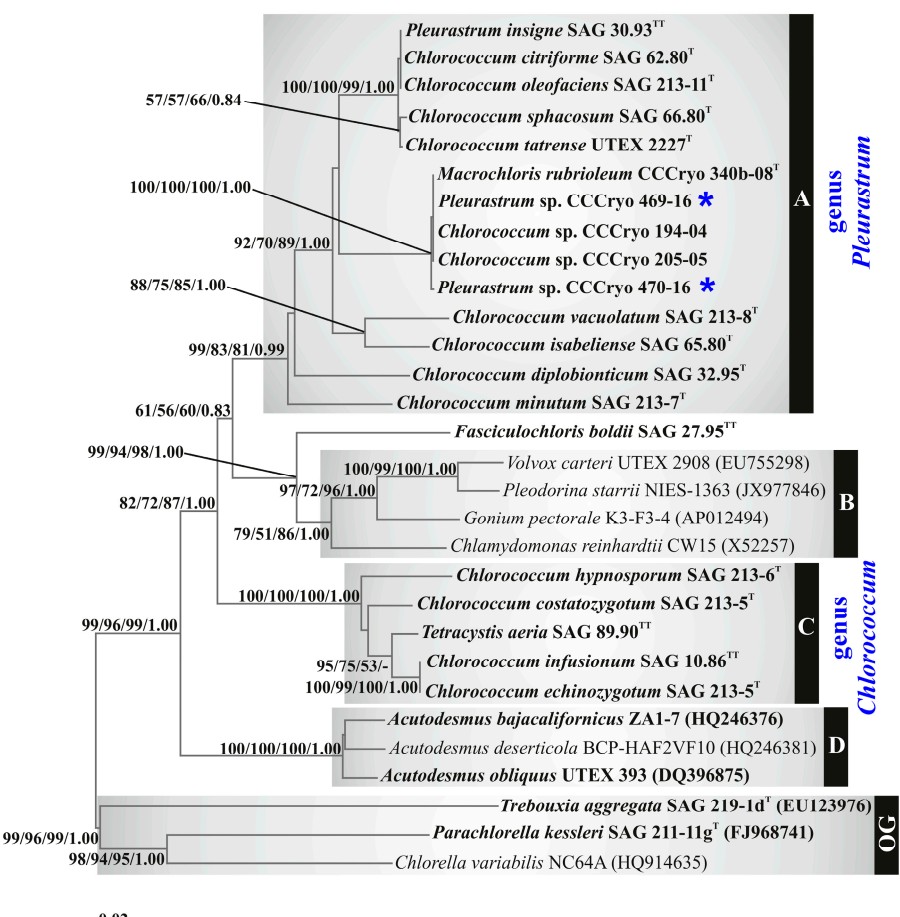

**Figure 7.** Phylogenetic reconstruction based on the *tuf* A gene fragment. Numbers associated with nodes indicate support values for NJ, MP, ML, and BI analyses, respectively. Only bootstrap supports ≥50% and posterior probabilities ≥0.70 are reported. Values for nodes that obtained support in only one of the performed phylogenetic analyses were omitted. The horizontal bar represents the expected number of nucleotide substitutions per site. Gray boxes represent the detected clades, tagged on the right with a capital letter from A to D; "OG" stands for "outgroup". Strains common to all the phylogenetic reconstructions are highlighted in boldface font. The superscript letter "T" near a strain name means that it is the authentic strain of the corresponding species; a double superscript letter "T" means that it is the authentic strain of a type species. The two new Antarctic isolates are indicated with a blue asterisk. The genera *Pleurastrum* and *Chlorococcum*, as circumscribed in this study, are also indicated.

Clade A (Figure 7) was composed of the Antarctic strains CCCryo 469-16 and CCCryo 470-16, the Arctic strains *Chlorococcum* sp. CCCryo 194-04 and *Chlorococcum* sp. CCCryo 205-05, *P. insigne* SAG 30.93, *M. rubrioleum* CCCryo 340b-08, *C. citriforme* SAG 62.80, *C. oleofaciens* SAG 213-11, *C. sphacosum* SAG 66.80, *C. tatrense* UTEX 2227, *C. vacuolatum* SAG 213-8, *C. isabeliense* SAG 65.80, *C. diplobionticum* SAG 32.95, and *C. minutum* SAG 213-7.

Inside clade A (Figure 7), six lineages could be detected: (1) *P. insigne* SAG 30.93, *C. citriforme* SAG 62.80, *C. oleofaciens* SAG 213-11, *C. sphacosum* SAG 66.80, and *C. tatrense* UTEX 2227 (clade support values: 100/100/99/1.00); (2) *M. rubrioleum* CCCryo 340b-08, the Antarctic strains CCCryo 469-16 and CCCryo 470-16, and the Arctic strains CCCryo 194-04 and CCCryo 205-05 (clade support values: 100/100/100/1.00); (3) *C. vacuolatum* SAG 213-8; (4) *C. isabeliense* SAG 65.80; (5) *C. diplobionticum* SAG 32.95; and (6) *C. minutum* SAG 213-7.

Clade C (Figure 7) was represented by *C. infusionum* SAG 10.86, *C. echinozygotum* SAG 213-5, *C. costatozygotum* SAG 20.95, *C. hypnosporum* SAG 213-6, and *T. aeria* SAG 89.80, with

again the highly supported cluster (clade support values: 100/99/100/1.00) formed by *C. infusionum* SAG 10.86 and *C. echinozygotum* SAG 213-5.

The phylogenetic reconstruction based on the ITS region (Figure 8) included the sequences of the two Antarctic isolates and of all, except one (i.e., *T. aeria* SAG 89.90), the strains obtained from international culture collections for comparison. The focal strains (i.e., the two Antarctic isolates and the strains obtained from international culture collections for comparison) were scattered between two well-supported clades, tagged as A (support values: 99/87/76/0.98) and E (support values: 99/99/95/1.00).

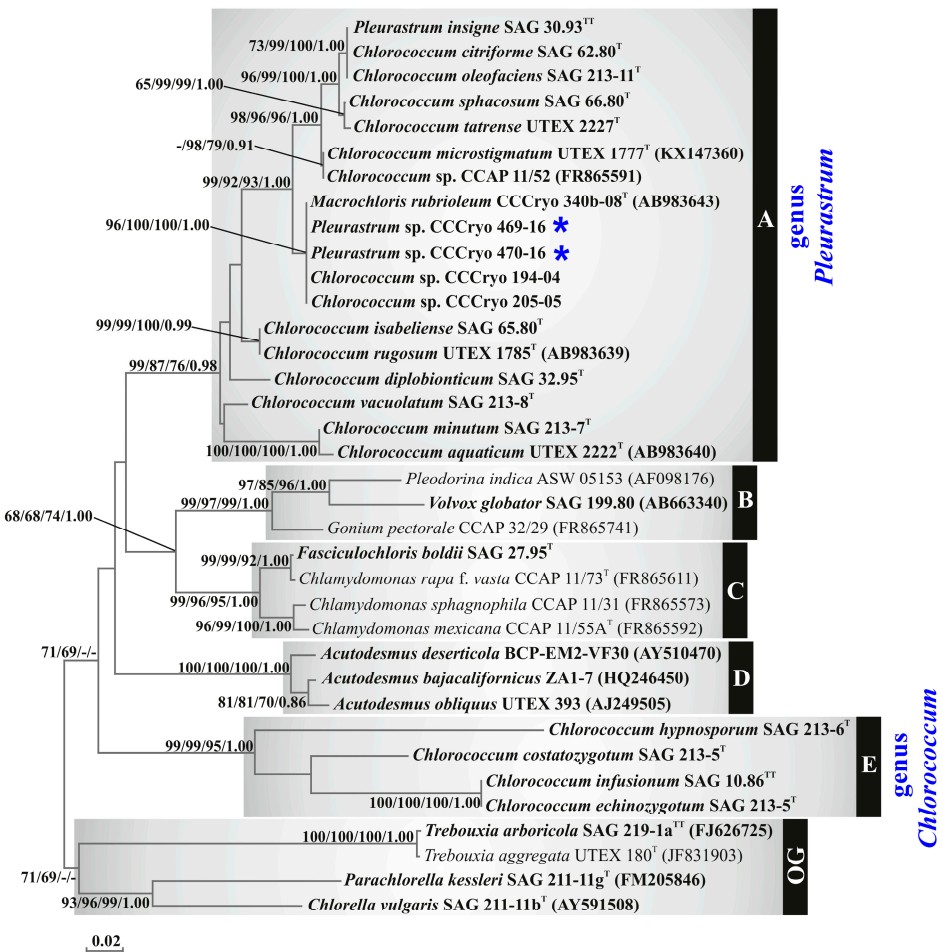

**Figure 8.** Phylogenetic reconstruction based on the ITS region. Numbers associated with nodes indicate support values for NJ, MP, ML, and BI analyses. Only bootstrap supports ≥50% and posterior probabilities ≥0.70 are reported. Values for nodes that obtained support in only one of the performed phylogenetic analyses were omitted. The horizontal bar represents the expected number of nucleotide substitutions per site. Gray boxes represent the detected clades, tagged on the right with a capital letter from A to E; "OG" stands for "outgroup". Strains common to all the phylogenetic reconstructions are highlighted in boldface font. The superscript letter "T" near a strain name means that it is the authentic strain of the corresponding species; a double superscript letter "T" means that it is the authentic strain of a type species. The two new Antarctic isolates are indicated with a blue asterisk. The genera *Pleurastrum* and *Chlorococcum*, as circumscribed in this study, are also indicated.

Clade A (Figure 8) was made up of the Antarctic strains CCCryo 469-16 and CCCryo 470-16, the Arctic strains CCCryo 194-04 and CCCryo 205-05, *M. rubrioleum* CCCryo 340b-08, *P. insigne* SAG 30.93, *C. citriforme* SAG 62.80, *C. oleofaciens* SAG 213-11, *C. sphacosum* SAG 66.80, *C. tatrense* UTEX 2227, *C. microstigmatum* UTEX 1777, *Chlorococcum* sp. CCAP 11/52, *C. isabeliense* SAG 65.80, *C. rugosum* UTEX 1785, *C. diplobionticum* SAG 32.95, *C. vacuolatum* SAG 213-8, *C. minutum* SAG 213-7, and *C. aquaticum* UTEX 2222.

Inside clade A (Figure 8), seven lineages were detected: (1) *P. insigne* SAG 30.93, *C. citriforme* SAG 62.80, *C. oleofaciens* SAG 213-11, *C. sphacosum* SAG 66.80, and *C. tatrense* UTEX 2227 (clade support values: 96/99/100/1.00); (2) *C. microstigmatum* UTEX 1777 and *Chlorococcum* sp. CCAP 11/52; (3) *M. rubrioleum* CCCryo 340b-08, the Antarctic strains CCCryo 469-16 and CCCryo 470-16, and the Arctic strains CCCryo 194-04 and CCCryo 205-05 (clade support values: 96/100/100/1.00); (4) *C. isabeliense* SAG 65.80 and *C. rugosum* UTEX 1785 (clade support values: 99/99/100/0.99); (5) *C. diplobionticum* SAG 32.95; (6) *C. vacuolatum* SAG 213-8; and (7) *C. minutum* SAG 213-7 and *C. aquaticum* UTEX 2222 (clade support values: 100/100/100/1.00).

Clade E (Figure 8) included *C. infusionum* SAG 10.86, *C. echinozygotum* SAG 213-5, *C. costatozygotum* SAG 20.95, and *C. hypnosporum* SAG 213-6; again, *C. infusionum* SAG 10.86 and *C. echinozygotum* SAG 213-5 were sister taxa with high support values (100/100/100/1.00).

### 3.4. Ultrastructural Observations on the Strains Obtained from International Culture Collections

For most of the focal strains included in clade A in all the phylogenetic reconstructions and for strains *C. infusionum* SAG 10.86 and *T. aeria* SAG 89.90, which were included together in a clade different from clade A in all the phylogenetic trees (i.e., clade B in the 18S rRNA tree, clade E in the *rbc*L and ITS region trees, and clade C in the *tuf*A tree), ultrastructural observations were carried out with a transmission electron microscope (Figures 9 and 10).

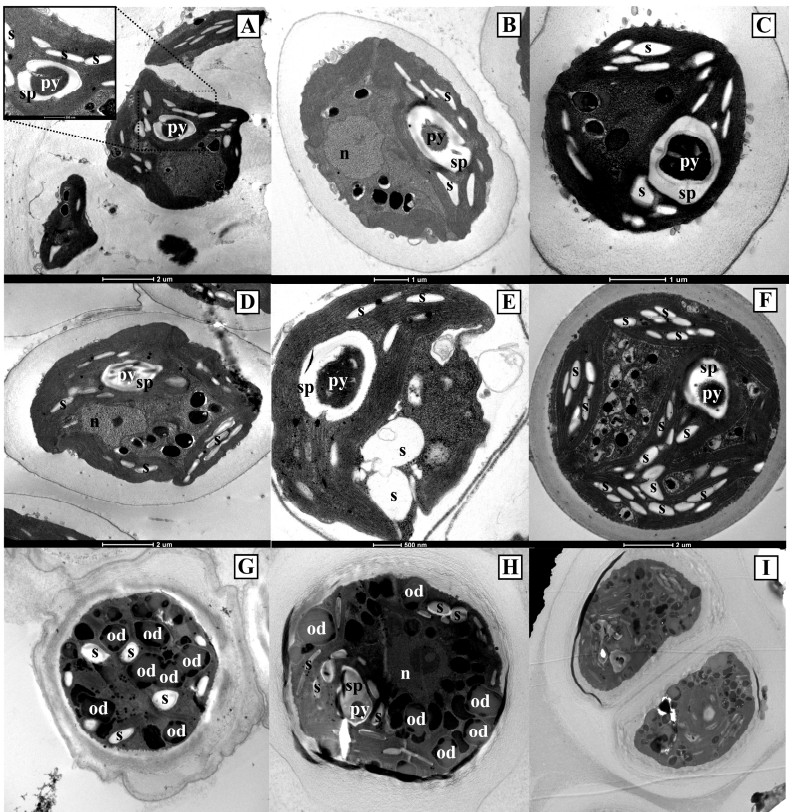

**Figure 9.** Transmission electron microscope micrographs of different focal microalgal strains included in clade A in the phylogenetic reconstructions. (**A**) *Pleurastrum insigne* SAG 30.93, (**B**) *Chlorococcum citriforme* SAG 62.80, (**C**) *Chlorococcum sphacosum* SAG 66.80, (**D**) *Chlorococcum isabeliense* SAG 65.80, (**E**) *Chlorococcum minutum* SAG 213-7, (**F**) *Chlorococcum vacuolatum* 213-8, (**G**) *Chlorococcum* sp. CCCryo 194-04, (**H**) *Chlorococcum* sp. CCCryo 205-05, and (**I**) a dyad of *Chlorococcum* sp. CCCryo 205-05. py = pyrenoid; sp = starch plate; s = starch; n = nucleus; od = oil droplet.

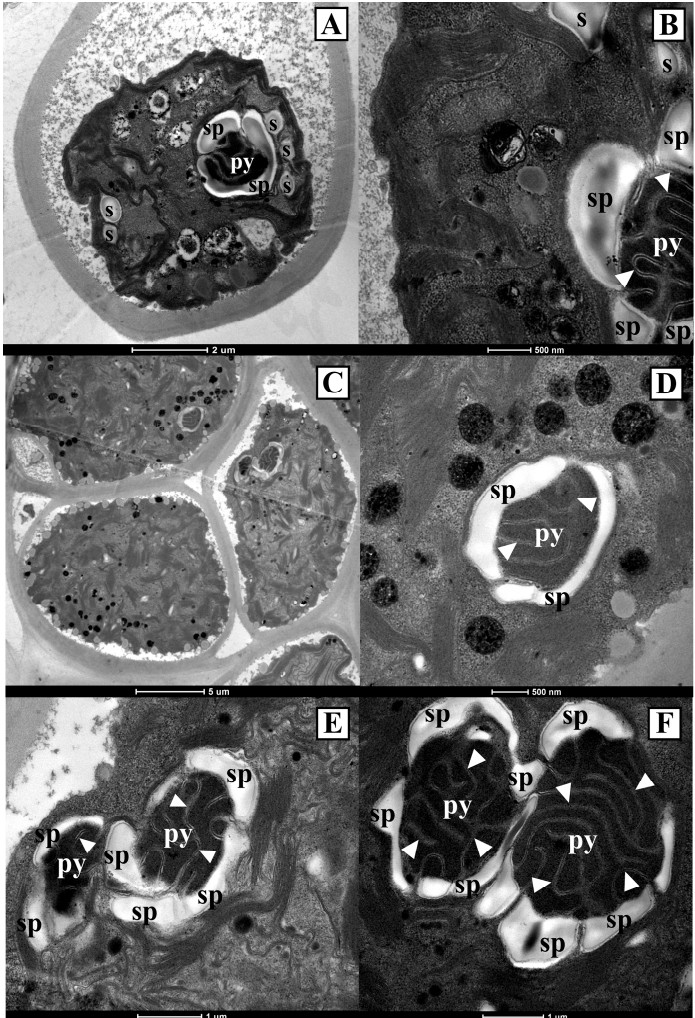

**Figure 10.** Transmission electron microscope micrographs of *Chloroccum infusionum* SAG 10.86 and *Tetracystis aeria* SAG 89.90. (**A**) *C. infusionum* SAG 10.86, (**B**) detail of pyrenoid *of C. infusionum*, (**C**) triad of *T. aeria* SAG 89.90, (**D**) detail of two pyrenoids of *T. aeria* SAG 89.90, (**E**) detail of two pyrenoids of *T. aeria* SAG 89.90, and (**F**) detail of pyrenoid of *T. aeria* SAG 89.90. py = pyrenoid; sp = starch plate; s = starch; white arrowheads = thylakoids crossing the pyrenoid.

    The strains included in clade A in all the phylogenetic reconstructions had a parietal chloroplast with several starch granules inside and a single pyrenoid (Figure 9). In all of the strains, the single pyrenoid was surrounded by a continuous starch plate, not crossed by thylakoids. The two Arctic strains CCCryo 194-04 and CCCryo 195-05, which were kept at 4 °C, were rich in oil droplets (Figure 9G,H).

    Strains *C. infusionum* SAG 10.86 and *T. aeria* SAG 89.90, belonging together to a clade different from clade A in all the phylogenetic trees, showed chloroplasts with one or two pyrenoids (Figure 10). In particular, *C. infusionum* SAG 10.86 cells had one pyrenoid in their chloroplast (Figure 10A,B), while *T. aeria* SAG 89.90 cells had one (Figure 10D) or two (Figure 10E,F) pyrenoids. Both for *C. infusionum* SAG 10.86 and *T. aeria* SAG 89.90, each pyrenoid was surrounded by a fragmented starch plate, and the pyrenoid was crossed by thylakoid bundles (Figure 10).

### 3.5. ITS2 Secondary Structure Results

    For 18 strains that were placed inside clade A (*Pleurastrum* genus) in the 18S rRNA, *rbc*L, *tuf*A, and ITS region phylogenetic reconstructions, the hypothetical secondary structures of the ITS2 portion, inside the ITS region, were predicted. The ITS2 sequences and secondary structures were used to generate a sequence + secondary structure multi-

alignment (Figure S2). From this multi-alignment, the ITS2 consensus secondary structure of the genus *Pleurastrum* was inferred (Figure S3). It was composed of four main helices, of which helix III was the longest (92–105 bases), followed by helix II (55–71 bases), helix I (35–38 bases), and helix IV (14–19 bases). Helix III was unbranched. Helices I, II, and III were similarly conserved among the *Pleurastrum* strains, with the basal portion of each helix the most conserved and the final portion the most variable; helix IV was instead highly variable. Five spacers were present, separating the helices and the 5.8S/LSU stem from each other; spacer 5, located between helix IV and the basal stem, was the longest (eight to 19 bases), while spacer 2, placed between helix I and helix II, was the shortest (three bases). Spacers 1, 2, and 3 were the most conserved, while spacers 4 and 5 were the most variable. In helix II, a uracil mismatch (U-U), classically found in this helix and falling inside a RNA processing site, was observed. The 5′ side of helix III was rich in the purine guanidine (G), and toward the helix tip, inside another RNA processing site, the highly conserved motif UGGGU was observed.

Following the method proposed by Darienko and collaborators [50] and with the help of the ITS2 sequence + secondary structure multi-alignment (Figure S2), in the paired portions of the *Pleurastrum* ITS2 consensus secondary structure, 87 barcode sites were detected and, successively, aligned (Figure S4), obtaining a numeric barcode alignment (Figure S5). The ITS2 sequence + secondary structure multi-alignment (Figure S2) and the ITS numeric barcode alignment (Figure S5) were subjected to phylogenetic analyses that led to a tree (Figure 11) representing the phylogenetic relationships among strains and species included in the genus *Pleurastrum*. Different species delimitation tests were applied on the ITS2 multi-alignment to help circumscribe the different lineages inside the genus *Pleurastrum*.

In the ITS2 phylogenetic reconstruction (Figure 11), eight lineages could be observed: (1) *P. insigne* SAG 30.93, *C. oleofaciens* SAG 213-11, *C. citriforme* SAG 62.80, *C. sphacosum* SAG 66.80, and *C. tatrense* UTEX 2227 (blue colored clade; support values: 99/99/99/81); (2) *C. microstigmatum* UTEX 1777 and *Chlorococcum* sp. CCAP 11/52 (turquoise blue colored clade; support values: 93/95/98/87); (3) *M. rubrioleum* CCCryo 340b-08, the Antarctic strains CCCryo 469-16 and CCCryo 470-16, and the Arctic strains CCCryo 194-04 and CCCryo 205-05 (gray colored clade; support values: 100/100/100/99); (4) *C. diplobionticum* SAG 32.95 (light green colored clade); (5) *C. isabeliense* SAG 65.80 and *C. rugosum* UTEX 1785 (red colored clade; support values: 97/100/100/88); (6) *C. vacuolatum* SAG 213-8 (brown colored clade); (7) *C. aquaticum* UTEX 2222 (yellow colored clade); and (8) *C. minutum* SAG 213-7 (orange colored clade). Indeed, *C. aquaticum* UTEX 2222 and *C. minutum* SAG 213-7 were sister taxa with high support values (99/100/100/100), but all the species delimitation tests detected them as separate lineages. Moreover, inside the blue colored clade, two sub-clades were evident: the one made up of *P. insigne* SAG 30.93, *C. oleofaciens* SAG 213-11, and *C. citriforme* SAG 62.80 (support values: 99/99/99/63) and the one constituted by *C. sphacosum* SAG 66.80 and *C. tatrense* UTEX 2227 (support values: 96/98/94/86). Two of the performed species delimitation tests (i.e., ABGD and PTP) detected the blue colored clade as a unique group, while the other two species delimitation tests (i.e., ASAP and GMYC) divided the blue colored clade into the two sub-clades just described. More in detail, taking into account all the ten partitions found with the ASAP test (Figure S6), only in one of these partitions was the blue colored clade considered as a unique group, while in four of them, only the group made up of *P. insigne* SAG 30.93, *C. oleofaciens* SAG 213-11, and *C. citriforme* SAG 62.80 was considered a unique lineage (two of these four partitions grouped *C. sphacosum* SAG 66.80 and *C. tatrense* UTEX 2227, and the other two indicated them as separate lineages).

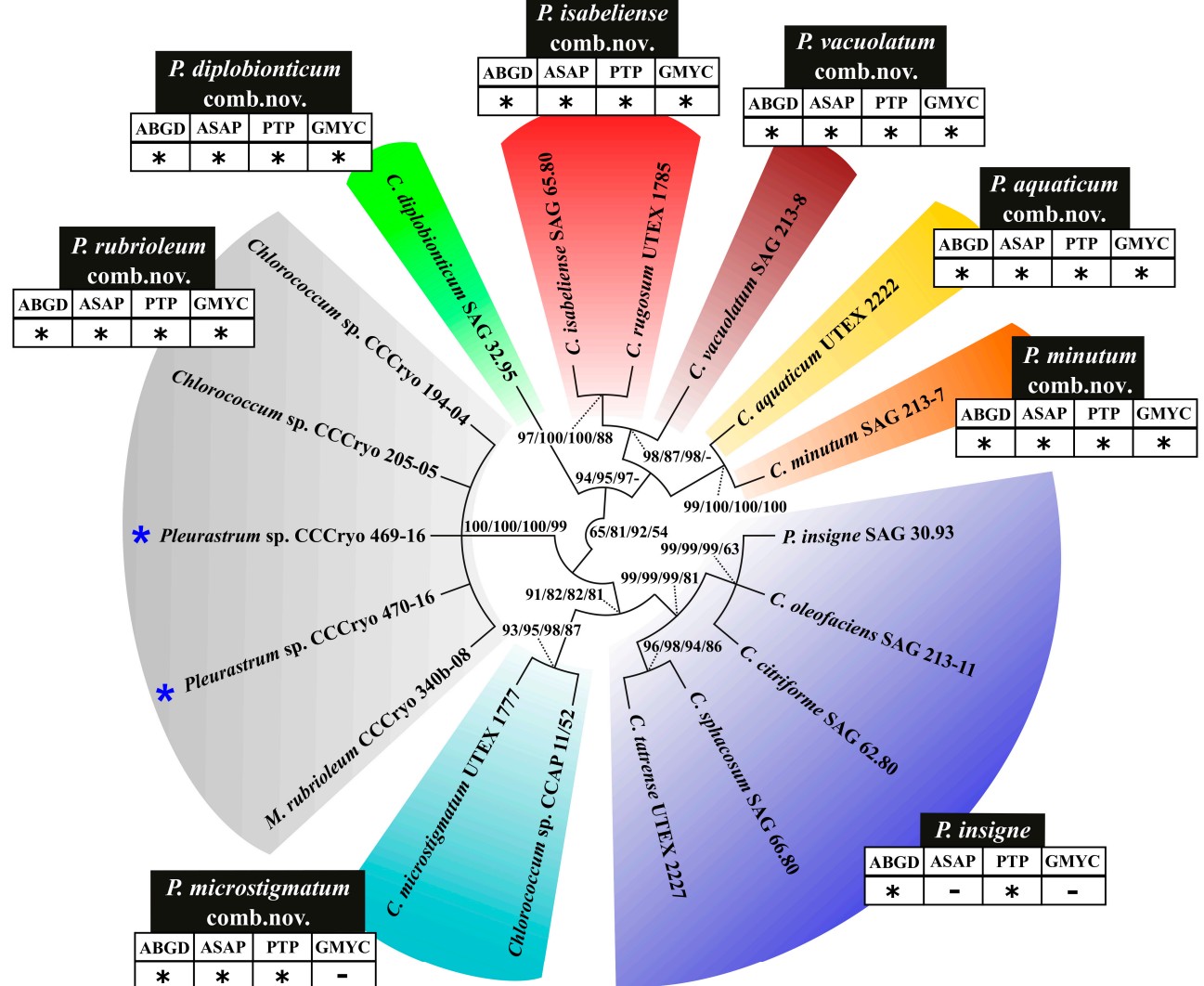

**Figure 11.** ITS2 tree illustrating the phylogenetic relationships among *Pleurastrum* strains and species. Numbers associated with nodes indicate support values for NJ analyses on the sequence + secondary structure multi-alignment, MP analyses on the only sequence multi-alignment, ML analyses on the only sequence multi-alignment, and MP analyses on the numeric barcode alignment. Only bootstrap supports ≥50% are reported. Values for nodes that obtained support in only one of the performed phylogenetic analyses were omitted. Colored boxes represent the different *Pleurastrum* species, and near each species clade, a black box with the species name (updated according to the results of this study) and a table reporting the species delimitation test results (an asterisk indicates that the test detected the clade, and a dash indicates that the test did not detect the clade) are depicted. The two new Antarctic isolates are indicated with a blue asterisk.

The ITS2 sequence + secondary structure multi-alignment was also used to detect the compensatory base changes (CBCs) and hemi-compensatory base changes (hCBCs) among the *Pleurastrum* ITS2 secondary structures (Figure 12).

No ITS2 CBCs were observed among the strains inside each of the colored clades highlighted in Figure 13. In addition, no hCBCs could be observed among the strains inside each colored clade of the ITS2 tree, with two exceptions: two hCBCs were observed between *C. microstigmatum* UTEX 1777 and *Chlorococcum* sp. CCAP 11/52 (turquoise blue colored clade), and one to five hCBCs were present among the strains included in the blue colored clade. More in detail, no hCBCs were observed among *P. insigne* SAG 30.93, *C. oleofaciens* SAG 213-11, and *C. citriforme* SAG 62.80; one hCBC was present between *C. sphacosum* SAG

66.80 and *C. tatrense* UTEX 2227; four hCBCs distinguished *C. sphacosum* SAG 66.80 from *P. insigne* SAG 30.93, *C. oleofaciens* SAG 213-11, and *C. citriforme* SAG 62.80; and five hCBCs separated *C. tatrense* UTEX 2227 from *P. insigne* SAG 30.93, *C. oleofaciens* SAG 213-11, and *C. citriforme* SAG 62.80 (Figure 12).

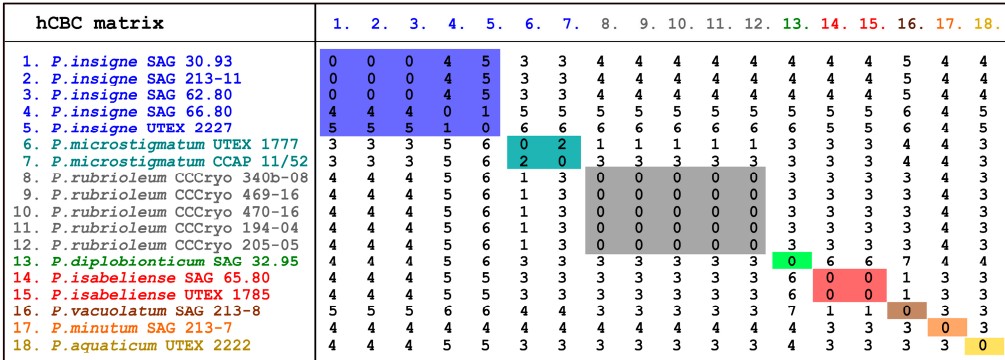

**Figure 12.** Matrices showing the compensatory base changes (CBCs) and hemi-compensatory base changes (hCBCs) among *Pleurastrum* ITS2 secondary structures. Strain species names are updated according to the results of this study and are highlighted by different colors.

The strains belonging to the genus *Pleurastrum*, as circumscribed in this study, were mapped onto a world map (Figure 13), updated by Peel and collaborators [60] with the climate zones according to the Köppen–Geiger climate classification. This investigation revealed a correlation of the ITS2 phylogenetic lineages highlighted in Figure 11, as well as the CBC data (Figure 12), with the different climate zones.

Based on the taxonomic priorities of the different species names (better discussed below), the blue colored clade in the ITS2 tree was attributed to the species *P. insigne*, the turquoise blue colored clade was assigned to the species *P. microstigmatum* comb. nov. (see the taxonomic treatment below), the gray colored clade was named as *P. rubrioleum* comb. nov. (see the taxonomic treatment below), and the red colored clade was assigned to the species *P. isabeliense* comb. nov. (see the taxonomic treatment below). The lineages represented by a single microalgal strain took the following species names: *P. diplobionticum* comb. nov., *P. vacuolatum* comb. nov., *P. aquaticum* comb. nov., and *P. minutum* comb. nov. (see the taxonomic treatment below) (Figure 11 and Table 2).

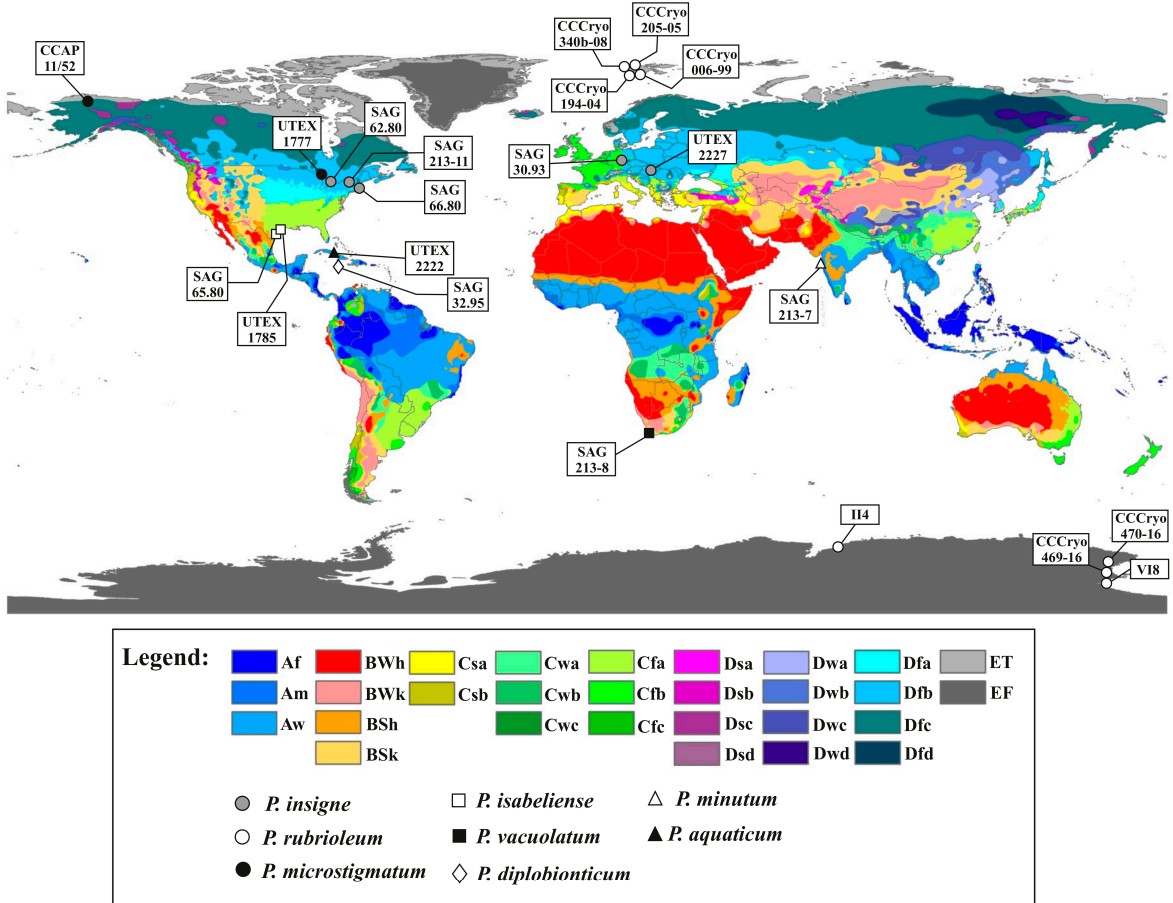

**Figure 13.** World map with the sampling localities from which the *Pleurastrum* strains considered in this study were sampled. The Köppen–Geiger climate zones are reported on the map as updated by Peel et al. [60]. Dfb = cold, without a dry season, warm summer; Dfc = cold, without a dry season, cold summer; ET = polar, tundra; EF = polar, frost; Cfa = temperate, without a dry season, hot summer; Csb = temperate, dry summer, warm summer; Am = tropical, monsoon; Aw = tropical, savannah; Af = tropical, rainforest. The species names of the different strains were updated according to the findings of this study.

**Table 2.** Updated species names and strains belonging to the genus *Pleurastrum* Chodat based on the present study. The species are listed in an order that mirrors the phylogenetic reconstructions. The authentic strains are highlighted by a superscript "T" near the strain identifier. For each strain, the geographical data (collection source, if available, and collection site) and the corresponding Köppen–Geiger climate zone are reported. The world Köppen–Geiger climate zones were retrieved from the work by Peel et al. [60]; Dfb = cold, without a dry season, warm summer; Dfc = cold, without a dry season, cold summer; ET = polar, tundra; EF = polar, frost; Cfa = temperate, without a dry season, hot summer; Csb = temperate, dry summer, warm summer; Am = tropical, monsoon; Aw = tropical, savannah; Af = tropical, rainforest.

| Updated Species Name | Strain Identifier | Geographical Data | Climate Zone |
|---|---|---|---|
| *Pleurastrum insigne* Chodat | SAG 30.93[T] | Soil from banks of brooklet Kahlquelle, Biebergemuend/Spessart, Hessen, Germany | Dfb |
| | SAG 62.80[T] | Soil from peat bog near Elkart, IN, USA | Dfb |
| | SAG 66.80[T] | Soil from *Sphagnum* bog, Falmouth, MA, USA | Dfb |
| | SAG 213-11[T] | Field soil at Duanesburg, NY, USA | Dfb |
| | UTEX 2227[T] | Snow detritus, Belianske Tatras (Belianske Tatry) mountains, Slovakia | Dfb |

**Table 2.** *Cont.*

| Updated Species Name | Strain Identifier | Geographical Data | Climate Zone |
|---|---|---|---|
| *Pleurastrum microstigmatum* (P.A. Archibald & Bold K. Sciuto, M.A. Wolf, M. Mistri & I. Moro comb. nov. | UTEX 1777[T] | Peat bog at Elkhart, IN, USA | Dfc |
| | CCAP 11/52 | Snow at Port Barrow, AK, USA | Dfc |
| *Pleurastrum rubrioleum* (Yur. Kawasaki & Nakada) K. Sciuto, M.A. Wolf, M. Mistri & I. Moro comb. nov. | CCCryo 340b-08[T] | Plastic bin/sewage cover in the vicinity west of the "Blaues Haus" in Ny-Ålesund, Spitsbergen, Svalbard, Norway | ET |
| | CCCryo 006-99 | East end of Zeppelinhamna near Ny-Ålesund, Spitsbergen, Svalbard, Norway | ET |
| | CCCryo 194-04 | Plastic bin/sewage cover in the vicinity west of the "Blaues Haus" in Ny-Ålesund, Spitsbergen, Svalbard, Norway | ET |
| | CCCryo 205-05 | Waste water runoff, Longyearbyen,, Spitsbergen, Svalbard, Norway | ET |
| | CCCryo 469-16 | Water pond near the OASI telescope hill, Mario Zucchelli Station, Terra Nova Bay, Victoria Land, Antarctica | EF |
| | CCCryo 470-16 | Snow at Edmonson Point, Wood Bay, Victoria Land, Antarctica | EF |
| | II4 | Ace Lake, Vestfold Hills, Antarctica | EF |
| | VI8 | Lake Fryxell, McMurdo Dry Valleys, Antarctica | EF |
| *Pleurastrum isabeliense* (P.A. Archibald & Bold) K. Sciuto, M.A. Wolf, M. Mistri & I. Moro comb. nov. | SAG 65.80[T] | Coastal sand, Port Isabel, TX, USA | Cfa |
| | UTEX 1785[T] | Coastal sand, Port Isabel, TX, USA | Cfa |
| *Pleurastrum vacuolatum* (R.C. Starr) K. Sciuto, M.A. Wolf, M. Mistri & I. Moro comb. nov. | SAG 213-8[T] | Soil from Cape Flats, South Africa | Csb |
| *Pleurastrum minutum* (R.C. Starr) K. Sciuto, M.A. Wolf, M. Mistri & I. Moro comb. nov. | SAG 213-7[T] | Soil from Mumbai, India | Am |
| *Pleurastrum aquaticum* (Archibald) K. Sciuto, M.A. Wolf, M. Mistri & I. Moro comb. nov. | UTEX 2222[T] | Treasure Lake (Lake El Tesero), Zapata, Cuba | Aw |
| *Pleurastrum diplobionticum* (Herndon) K. Sciuto, M.A. Wolf, M. Mistri & I. Moro comb. nov. | SAG 32.95[T] | Soil from corn field near Daniel Town, Jamaica | Af |

The ITS2 haplotypes study (Figure 14) allowed the evaluation of the differences between each *Pleurastrum* ITS2 haplotype and putting the different haplotypes in relation with the species found with the ITS2 phylogenetic analyses (Figure 11) and the climate zones where the different strains were isolated (Figure 13). Three ITS2 haplotypes were detected for *P. insigne*, with haplotypes 2 and 3 being very similar (one nucleotide difference) and haplotype 1 being more distant from the other two (seven to eight nucleotide differences); all the haplotypes were sampled from the Cold-Dfb climate zone (i.e., cold, without a dry season, with a warm summer) (Figures 13 and 14). Two haplotypes (haplotypes 4 and 5) were found for the *P. microstigmatum* comb. nov., differing for five nucleotide substitutions and both isolated from the Cold-Dfc climate zone (i.e., cold, without a dry season, with a cold summer) (Figures 13 and 14). Just one haplotype (haplotype 6) was detected for the new species *P. rubrioleum* comb. nov., whose strains were all sampled from the Polar-ET (i.e., polar, tundra) and Polar-EF (i.e., polar, frost) climate zones (Figures 13 and 14). Haplotype 7 corresponded to the species *P. diplobionticum* comb. nov., sampled from the Tropical-Af climate zone (i.e., tropical, rainforest) (Figures 13 and 14). Just one haplotype (haplotype 8) was found for *P. isabeliense* comb. nov., and both the strains belonging to this species were isolated from the Temperate-Cfa climate zone (i.e., temperate, without a dry season, with a hot summer) (Figures 13 and 14). Haplotype 9 corresponded to *P. vacuolatum* comb. nov., a species from the Temperate-Csb climate zone (i.e., temperate, with a dry and warm summer) (Figures 13 and 14). Haplotype 10 represented the species *P. aquaticum* comb. nov., found in the Tropical-Aw climate zone (i.e., tropical, savannah) (Figures 13 and 14). Haplotype 11 corresponded to *P. minutum* comb. nov., isolated from the Tropical-Am climate zone (i.e., tropical, monsoon) (Figures 13 and 14).

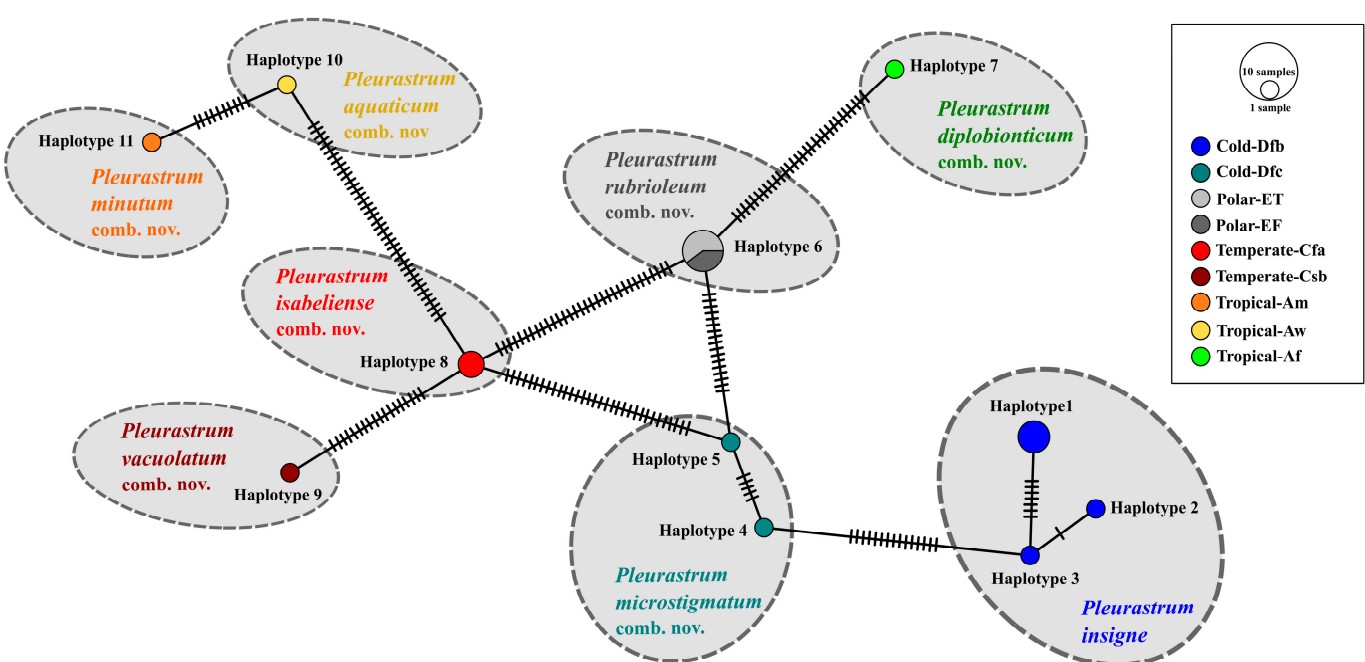

**Figure 14.** Minimum spanning network of the ITS2 haplotypes found for the genus *Pleurastrum* in relation to the different climate zones. Dfb = cold, without a dry season, warm summer; Dfc = cold, without a dry season, cold summer; ET = polar, tundra; EF = polar, frost; Cfa = temperate, without a dry season, hot summer; Csb = temperate, dry summer, warm summer; Am = tropical, monsoon; Aw = tropical, savannah; Af = tropical, rainforest. The species names attributed to the different haplotype groups were updated according to the findings of this study.

## 4. Discussion

During sampling campaigns aimed at screening the Antarctic microflora biodiversity, two green coccoid microalgae were isolated from the Victoria Land region (Antarctica). A first phylogenetic analysis, based on the 18S rRNA gene, showed that the two Antarctic isolates (tagged as CCCryo 469-16 and CCCryo 470-16) were related to strains attributed the genera *Chlorococcum* Meneghini, *Macrochloris* Korshikov, and *Pleurastrum* Chodat. *Chlorococcum*, first described by Meneghini in 1843 [61], is a wide genus of coccoid microalgae, including 48 accepted species names and considered to be cosmopolitan [19]. *Macrochloris* was described by Korshikov in 1926 [62] and comprises six currently accepted species, all composed of unicellular microalgae [19]. The taxonomical history of the genus *Pleurastrum* is complicated, and since the description of this highly polymorphic taxon by Chodat in 1894 [16], it has been subject to several revisions that led to just three to four taxonomically accepted species [17,19]. Recently, based on 18S rRNA analyses, Kawasaki and collaborators [20] showed that the authentic strain SAG 30.93 of *P. insigne* (i.e., the type species of the genus *Pleurastrum*) [18] genetically coincided with *C. oleofaciens* SAG 213-11 (authentic strain of *C. oleofaciens*). This and the failed observation of a filamentous form for strain SAG 30.93, kept at the same culture conditions reported by Sluiman and Gärtner [18], led Kawasakii and collaborators [20] to conclude that strain SAG 30.93 had been overgrown by a *C. oleofaciens* strain and, thus, to transfer strain SAG 30.93 under the genus *Chlorococcum*. Nevertheless, Kawasakii et al. [20] did not include the authentic strain of the type species of *Chlorococcum*, *Chlorococcum infusionum* (Schrank) Meneghini (i.e., strain SAG 10.86), in their phylogeny; this would have indicated the true *Chlorococcum* clade. Moreover, we have accurately re-examined the original description and drawings of *P. insigne* by Chodat [16], and indeed, we did not find any explicit reference to the presence of filamentous forms. In particular, neither Chodat's original description of *P. insigne* nor his description of the entire genus *Pleurastrum* clearly reports the presence of filamentous phases:

"*Pleurastrum* nov. genus.

Algue unicellulaire se reproduisant par tétrades de cellules dans l'intérieur de la membrane primitive, munie à l'état parfait de sculptures sur la membrane, formant des tétrades compliquées pouvant se résoudre en états gleocystis et produisant des zoospores à deux cils.

*Pleurastrum insigne nob.*

Cette algue diffère à peine de la précédente, sinon par sa membrane souvent sculptée et par ses colonies gleocystoïdes se formant facilement, enfin par son chromatophore plus large, plus étalé. Il se pourrait cependant que l'espèce précédente doive rentrer dans ce genre".

Regarding the original Chodat's drawings of *P. insigne* (pl. 28: figs 16–45 in [16]), only drawing no. 45 resembles a filamentous form. However, we cannot say if it is a real filamentous phase, as later interpreted by all the following authors, or a filamentous-like cell assemblage.

The gross morphology of the two Antarctic isolates CCCryo 469-16 and CCCryo 470-16 can fit in with the description of all the cited genera (i.e., *Chlorococcum*, *Macrochloris*, and *Pleurastrum*), even if a short occasional filamentous-like phase (whether it was real filaments or not) and sarcinoid forms, typically attributed to the genus *Pleurastrum* [16,17], were observed only just after the isolation of the two strains (data not shown) and then were lost in the laboratory culture conditions. Indeed, it is known that microalgal morphology can be strongly influenced by environmental conditions and that several features, observed in natural populations just after their sampling, can become lost when in laboratory cultures over time, e.g., [63–66]. Instead, other morphological features reported by Chodat for the genus *Pleurastrum* are present in the Antarctic isolates, such as the reproduction through the formation of cell tetrads, inside the mother cell membrane (Figure 4C), which later divide further, giving rise to several cells inside the mother cell membrane (Figure 4D). Daughter cells are then released through the rupture of the mother cell membrane (Figure 3A). Zoospores were also observed (Figure 2E).

Therefore, besides the first analyses based on the 18S rRNA gene, further molecular markers (i.e., *rbc*L, *tuf*A, and ITS, with a focus on the ITS2 secondary structures) were employed to provide a more precise taxonomic identification to strains CCCryo 469-16 and CCCryo 470-16. To accomplish this scope, several Chlorophyta strains belonging to the genera *Chlorococcum*, *Macrochloris*, and *Pleurastrum* were obtained from international culture collections for comparison, mainly choosing the strains phylogenetically related to the Antarctic isolates according to the obtained 18S rRNA tree. Among the comparison strains, the former authentic strain of *P. insigne*, SAG 30.93, and the authentic strain of *C. infusionum*, SAG 10.86, were selected.

The choice of using several loci (i.e., multilocus approach) from different cell compartments was made to obtain stronger results. In particular, besides being among the most used molecular markers for green algal systematics, the selected loci belong to two different cell compartments: the nuclear genome (in the case of the 18S rRNA gene and the ITS region) and the plastid genome (in the case of the *rbc*L and *tuf*A genes). It is known that the nuclear and plastid genomes have different evolutionary histories, with the latter usually showing a slower evolution rate and therefore being more conserved [67]. As could be expected, the different evolution of the nuclear and plastid genomes led to different relationships among the most internal nodes of the obtained nuclear and plastid phylogenies. Nevertheless, at the terminal nodes, all the obtained phylogenies showed congruent results since the same clades were detected. As already highlighted by other authors, e.g., [66], the congruence between nuclear and plastid molecular evidence provides more reliability to the detected lineages, in particular in organisms whose conspecificity cannot be verified by evaluating their sexual compatibility (i.e., biological species concept).

In all the phylogenetic reconstructions obtained in this study, *P. insigne* SAG 30.93 and *C. infusionum* SAG 10.86 were placed into two distinct clades, which were well supported and clearly unrelated. Taking into account the above reported considerations about Chodat's original description [16] and also the possibility that certain features can become lost

when microalgal strains are cultivated for a long time, we suggest that strain SAG 30.93 still represents the reference strain for the species *P. insigne*. This and our phylogenetic findings allow us to state that the first clade (tagged as clade A in all the obtained trees; Figures 5–8), comprising *P. insigne* SAG 30.93, represents the genus *Pleurastrum*, and the second clade (i.e., clade B in the 18S rRNA tree (Figure 5), clade C in the *tuf*A tree (Figure 7), and clade E in both the *rbc*L and ITS trees (Figures 6 and 8, respectively)), containing *C. infusionum* SAG 10.86, represents the genus *Chlorococcum*.

At the ultrastructural level, strains belonging to the here circumscribed genus *Pleurastrum* are clearly distinguishable from the strains belonging to the genus *Chlorococcum*. Indeed, in the genus *Pleurastrum*, only one pyrenoid is found inside each chloroplast; moreover, the pyrenoid is not crossed by thylakoids, and it is surrounded by a continuous starch plate (Figure 9). Conversely, in the genus *Chlorococcum*, more than one pyrenoid can be present in each chloroplast, and each pyrenoid is crossed by thylakoid membranes and surrounded by a fragmented starch plate (Figure 10). This pyrenoid diagnostic feature for the real *Chlorococcum* members had already been reported by Watanabe and Lewis [68]. Moreover, the existence of this different pyrenoid morphology among the strains previously attributed to the genus *Chlorococcum* was noticed also by other authors [69,70].

Inside the genus *Chlorococcum*, the authentic strain *T. aeria* SAG 89.90 was included (Figures 5–7), thus confirming the recent attribution of this species to *Chlorococcum* as *C. aerium* [68]. In all the obtained phylogenies, the authentic strains of *C. infusionum* and *C. echinozygotum* were sister taxa with high support values (Figures 5–8); moreover, they showed a high sequence identity for all the selected markers (i.e., 99.77% for the 18S rRNA, 100% for the *rbc*L, 100% for the *tuf*A, and 100% for the ITS). This finding allows us to synonymize the species *C. echinozygotum* with *C. infusionum*, whose name has the taxonomic priority (see the taxonomic treatment below). Therefore, based on the present results, four species unequivocally belong to the genus *Chlorococcum*: *C. infusionum*, *C. costatozygotum*, *C. hypnosporum*, and *C. aerium*.

Taking the molecular results as a whole, inside the genus *Pleurastrum*, eight well-supported lineages were detected, with the inclusion under this taxon of several authentic strains of species previously attributed to *Chloroccoccum* and the authentic strain of *Macrochloris rubrioleum*, CCCryo 340b-08. In addition, the two Antarctic isolates CCCryo 469-16 and CCCryo 470-16 were placed in the genus *Pleurastrum* and, in particular, in the sub-clade including *M. rubrioleum* CCCryo 340b-08, collected from Norway (Arctic zone), three other strains from this country (CCCryo 194-04, CCCryo 205-05, and CCCryo 006-99; the last was present only in the 18S rRNA tree), and two other isolates from Antarctica (II4 and VI8; only in the 18S rRNA tree). No CBCs nor hCBCs were found between the ITS2 secondary structures of these Arctic and Antarctic strains, further suggesting that they belong to the same species. Indeed, it was demonstrated that when a CBC occurs between two members of the same genus, they belong to different species with a reliability of 93.11% [71]; the absence of CBCs among two organisms is not an unchallengeable proof of their conspecificity but strongly supports this hypothesis. This clade was confirmed also by all the performed species delimitation tests (i.e., ABGD, ASAP, PTP, and GMYC) and by the provenance of all the included strains from the Polar climate zone. Therefore, for this lineage, we here propose the species name *Pleurastrum rubrioleum* comb. nov. (see Table 2 and the taxonomic treatment below). Only one ITS2 haplotype was found for *P. rubrioleum*, with no distinction between strains from Arctic and Antarctic zones.

Five other *Pleurastrum* lineages, highlighted by the obtained molecular phylogenies and in particular by the ITS2 tree, were confirmed with all the applied species delimitation tests: *Pleurastrum isabeliense* comb. nov. (with no CBCs found between the ITS2 secondary structures of the two strains included in this sub-clade), *Pleurastrum vacuolatum* comb. nov., *Pleurastrum minutum* comb. nov., *Pleurastrum aquaticum* comb. nov. and *Pleurastrum diplobionticum* comb. nov. (see Table 2 and the taxonomic treatment below). These five species were each found in a different climate zone, and for each lineage, only one haplotype was found.

The GMCY test was the only one that did not support the conspecificity of strains UTEX 1777 and CCAP 11/52, which showed also one hCBC between their ITS2 secondary structures. However, the good statistical values of this lineage (present in the 18S rRNA, ITS, and ITS2 phylogenies), the absence of CBCs between their ITS2 secondary structures, and their provenance from the same climate zone (i.e., Dfc = cold, without a dry season, with a cold summer) strongly suggest that these two strains belong to the same species, for which we here propose the name *Pleurastrum microstigmatum* comb. nov. (see Table 2 and the taxonomic treatment below).

In addition, for the clade including *P. insigne* SAG 30.93 and four other authentic strains, formerly under the genus *Chlorococcum* (i.e., SAG 213-11, SAG 62.80, SAG 66.80, and UTEX 2227), two of the performed species delimitation tests (i.e., ASAP and GMYC) did not recognize this group as a unique clade; moreover, four to five hCBCs were found between strains SAG 66.80 and UTEX 2222, from one side, and strains SAG 30.93, SAG 213-11 and SAG 62.80, from the other. However, as in the case of the *P. microstigmatum* lineage, the five above cited strains were grouped with high statistical supports in all the obtained phylogenies, did not show CBCs among their ITS2 secondary structures, and belonged to the same climate zone (i.e., Dfb = cold, without a dry season, with a warm summer). In addition, this clade was also previously found by Kawasaki and collaborators [20]. Therefore, taking into account the taxonomic priorities, we recognize this clade as the true *Pleurastrum insigne* species (see Table 2 and the taxonomic treatment below).

In the case of both *P. insigne* and *P. microstigmatum*, more ITS2 haplotypes were found: three for the former (i.e., haplotypes 1, 2, and 3) and two for the latter species (i.e., haplotypes 4 and 5). *P. insigne* haplotype 1 was common to strains SAG 30.93, SAG 213-11, and SAG 62.80 and differed for six nucleotide substitutions from haplotype 2 (strain SAG 66.80) and haplotype 3 (strain UTEX 2227); haplotypes 2 and 3 were distinguished for just one nucleotide substitution. The haplotype differences mirrored the found hCBCs and the distinction inside *P. insigne* evidenced by the ASAP and GMYC species delimitation tests. The same was observed for *P. microstigmatum*, whose ITS2 haplotypes differed for five nucleotide substitutions.

In addition, the species *Chlorococcum ellipsoideum* Deason & Bold potentially belong to *Pleurastrum* based on the *rbc*L tree; in fact, in that phylogeny, the authentic strain of this species, UTEX 972, and *C. ellipsoideum* LUCC 005 are included under this genus. However, further evidence is needed to confirm this hypothesis.

Concluding, the results obtained in this work allow us to re-define the borders of the genus *Pleurastrum* and suggest the climate-driven differentiation of the species included under this taxon, in spite of their simple and overlapping morphologies, as well as the high phenotypic plasticity described previously [16,17]. The correlation between molecular lineages and ecological parameters has been already demonstrated for other algal taxa, e.g., [72–75]. This highlights the importance of an integrative taxonomy approach that takes into account other features than just the morphological and genetic aspects. In the future, more analyses on the investigated microalgal strains can reveal other characters, such as the biochemical/metabolic ones probably correlated to the climate zones, which could help differentiating the *Pleurastrum* species further.

Currently, the monophyletic species concept, as expressed by Johansen and Casamatta [76] and applied in this study, is probably the most reliable criterion to establish species boundaries for asexual organisms and/or for organisms whose sexual compatibility is in any way difficult to be assessed, such as most microalgal taxa. According to this concept, species are "monophyletic clusters of strains or natural populations that are diagnosable by some unique combination of traits, those traits being any combination of morphological, biochemical, molecular, or other characteristics" [76].

## 5. Conclusions

Based on the findings of this study, the following taxonomic treatment is proposed, with the taxonomic changes highlighted in boldface font:

*Chlorococcum infusionum* (Schrank) Meneghini 1842: 27, pl. 2: fig 3. Published in *Memorie della Reale Accademia delle Scienze di Torino, ser. 2,* 1843, 5: 1–143.

Type: Lectotype; designated by Silva and Starr 1953

Type locality: München, Germany (?)

Reference strain: SAG 10.86 (CAUP H 811)

Molecular vouchers: 18S rRNA gene: KM020174; *rbc*L gene: LT594548; *tuf* A gene: LT594534; ITS region: LT594565

**Heterotypic synonyms**: *Chantransia infusionum* Schrank 1814: 20; *Cystococcus humicola* Nägeli 1849: 85, pl. III: E; *Chlorococcum humicola* (Nägeli) Rabenhorst 1868: 58; *Gloeocystis humicola* (Nägeli) J.Adams 1908: 33; *Chlorococcum humicola* var. *incrassatum* F.E.Fritsch & R.P.John 1942: 377; *Chlorococcum echinozygotum* R.C.Starr 1955: 18

Other molecularly verified strains: SAG 213-5

Distribution: widely distributed in Europe, Asia, North America, South America, Australia, New Zealand, Arctic [19]

*Pleurastrum insigne* Chodat 1894: 613, pl. 28: figs 16–45. Published in *Bulletin de l'Herbier Boissier,* 1894, 2, 585–616.

Type: Lectotype; designated by Sluiman & Gärtner 1990, p. 133

Type locality: Carouge, Geneva, Switzerland

Reference strain: SAG 30.93

Molecular vouchers: 18S rRNA gene: AB983614; *rbc*L gene: EF113464; *tuf* A gene: LT594537; ITS region: LT594568

**Heterotypic synonyms:** *Chlorococcum oleofaciens* Trainor & Bold 1954: 759; *Chlorococcum citriforme* P.A. Archibald & Bold 1970: 24; *Chlorococcum sphacosum* P.A. Archibald & Bold 1970: 45; *Chlorococcum tatrense* P.A. Archibald 1979: 306

Other molecularly verified strains: SAG 213-11, SAG 62.80, SAG 66.80, UTEX 2227

Distribution: Britain and Ireland, Germany, Switzerland, India [19]; Slovakia, Indiana (USA), Massachusetts (USA), New York (USA) [based on this study]

*Pleurastrum microstigmatum* (P.A. Archibald & Bold) K. Sciuto, M.A. Wolf, M. Mistri & I. Moro **comb. nov.**

Basionym: *Chlorococcum microstigmatum* P.A. Archibald & Bold 1970: 34, figs 24, 63. Published in *Phycological studies XI. The genus Chlorococcum Meneghini. University of Texas Publication,* 1970, 7015, 1–115.

Type: Holotype; based on isolate 18T2A

Type locality: Elkhart, Indiana, USA

Reference strain: UTEX 1777

Molecular vouchers: 18S rRNA gene: AB983616; ITS region: KX147360

Other molecularly verified strains: CCAP 11/52

Distribution: Indiana (USA) [19]; Alaska (USA) [based on this study]

*Pleurastrum rubrioleum* (Yur. Kawasaki & Nakada) K. Sciuto, M.A. Wolf, M. Mistri & I. Moro **comb. nov.**

Basionym: *Macrochloris rubrioleum* Yur. Kawasaki & Nakada 2015: 1014–1015, Figure 10. Published in *Journal of Phycology*, 2015, *51*, 1000–1016.

Type: Holotype; TNS-AL-58908

Type locality: Ny-Ålesund, Spitsbergen, Svalbard, Norway

Reference strain: CCCryo 340b-08

Molecular vouchers: 18S rRNA gene: GU117573; *tuf* A gene: LT989896; ITS region: AB983643; *rbc*L gene sequence not available for the reference strain (ex-holotype), but for other four strains: LT594543, LT594544, LT594551, LT594552

Other molecularly verified strains: CCCryo 006-99, CCCryo 194-04, CCCryo 205-05, CCCryo 469-16, CCCryo 470-16, II4, VI8

Distribution: Artic (Norway) and Antarctica [based on this study]

*Pleurastrum isabeliense* (P.A. Archibald & Bold) K. Sciuto, M.A. Wolf, M. Mistri & I. Moro **comb. nov.**

Basionym: *Chlorococcum isabeliense* P.A. Archibald & Bold 1970: 31, figs 20, 58, 59. Published in *Phycological studies XI. The genus Chlorococcum Meneghini. University of Texas Publication*, 1970, 7015, 1–115.

Type: Lectotype; based on isolate P. I. 9-2

Type locality: Port Isabel, Texas (USA)

Reference strain: SAG 65.80 (UTEX 1774)

Molecular vouchers: 18S rRNA gene: KM020106; *tuf* A gene: LT594532; ITS region: LT594563

**Heterotypic synonyms:** *Chlorococcum lacustre* P.A. Archibald & Bold 1970: 32; ***Chlorococcum rugosum*** P.A. Archibald & Bold 1970: 43

Other molecularly verified strains: UTEX 1785

Distribution: Russia, Texas (USA), Indiana (USA), Hawaiian Islands (USA) [19]

***Pleurastrum vacuolatum*** (R.C. Starr) K. Sciuto, M.A. Wolf, M. Mistri & I. Moro **comb. nov.**

Basionym: *Chlorococcum vacuolatum* R.C. Starr 1954: 143, figs 7–10. Published in *Lloydia (Cincinnati)*, 1954, *16*, 142–148.

Type: Lectotype; Starr 1952

Type locality: Cape Flats, South Africa

Reference strain: SAG 213-8 (CCAP 213/8; UTEX 110)

Molecular vouchers: 18S rRNA gene: KM020107; *tuf* A gene: LT594529; ITS region: LT594560

Heterotypic synonyms: *Chlorococcum perforatum* Arce & Bold 1958: 497; *Chlorococcum loculatum* P.A. Archibald & Bold 1970: 33; *Chlorococcum salsugineum* P.A. Archibald & Bold 1970: 44; *Chlorococcum texanum* Archibald & Bold 1970: 46

Distribution: Italy, Russia, Cuba, Indiana (USA), Massachusetts (USA), Texas (USA) [19]

***Pleurastrum minutum*** (R.C. Starr) K. Sciuto, M.A. Wolf, M. Mistri & I. Moro **comb. nov.**

Basionym: *Chlorococcum minutum* R.C. Starr 1955: 30, figs 81–103. Published in *Indiana University Publications. Science Series*, 1955, 20: 1–111.

Type: Holotype; based on isolate Bold W7-2

Type locality: Mumbai, India

Reference strain: SAG 213-7 (UTEX 117)

Molecular vouchers: 18S rRNA gene: KM020099; *rbc*L gene: LT594546; *tuf* A gene: LT594528; ITS region: LT594559

Heterotypic synonyms: *Chlorococcum scabellum* Deason & Bold 1960: 23; *Chlorococcum aureum* P.A. Archibald & Bold 1970: 23; *Chlorococcum reticulatum* P.A. Archibald & Bold 1970: 42; *Chlorococcum typicum* P.A. Archibald & Bold 1970: 46

Distribution: widely distributed in Europe and North America, present in Argentina, India, Nepal [19]

***Pleurastrum aquaticum*** (P.A. Archibald) K. Sciuto, M.A. Wolf, M. Mistri & I. Moro **comb. nov.**

Basionym: *Chlorococcum aquaticum* P.A. Archibald 1979: 309, Figure 7. Published in *British Phycological Journal*, 1979, *14*, 305–312.

Type: Holotype; based on isolate KOM 64/88

Type locality: Lake El Tesero, Zapata, Cuba

Reference strain: UTEX 2222

Molecular vouchers: 18S rRNA gene: AB983622; ITS region: AB983640

Distribution: Cuba [19]

***Pleurastrum diplobionticum*** (Herndon) Sciuto, Wolf, Mistri & Moro **comb. nov.**

Basionym: *Chlorococcum diplobionticum* Herndon 1958: 308, figs 1–30. Published in *American Journal of Botany*, 1958, 45(4): 298–308.

Type: Holotype; xii.1949; based on isolate WH1-J

Type locality: near Daniel Town, Jamaica

Reference strain: SAG 32.95 (UTEX 950)

Molecular vouchers: *tuf* A gene: LT594536; ITS region: LT594567

Distribution: Jamaica, Russia [19]

**Supplementary Materials:** The following supporting information can be downloaded at https://www.mdpi.com/article/10.3390/d15050650/s1: Figure S1: Overview, using the light microscope, of most of the microalgal strains obtained from international culture collections for comparison; Figure S2: *Pleurastrum* ITS2 complete alignment; Figure S3: Hypothetical ITS2 consensus secondary structure of the genus *Pleurastrum*; Figure S4: *Pleurastrum* ITS2 barcode site alignment; Figure S5: ITS2 numeric barcode alignment of *Pleurastrum*, derived from the sequence + secondary structures multi-alignment of only the barcode sites; Figure S6: Complete results of the ASAP test on the ITS2 final alignment.

**Author Contributions:** Conceptualization, K.S., M.A.W. and I.M.; methodology, K.S., M.A.W. and I.M.; software, K.S. and M.A.W.; validation, K.S., M.A.W. and I.M.; formal analysis, K.S. and M.A.W.; investigation, K.S., M.A.W. and I.M.; resources, K.S. and I.M.; data curation, K.S. and M.A.W.; writing—original draft preparation, K.S. and M.A.W.; writing—review and editing, K.S., M.A.W., M.M. and I.M.; visualization, K.S.; supervision, K.S., M.A.W., M.M. and I.M.; project administration, K.S., M.A.W. and I.M.; funding acquisition, K.S. and I.M. All authors have read and agreed to the published version of the manuscript.

**Funding:** This research was funded by the National Antarctic Research Project (PNRA) 2004–2006. Katia Sciuto realized this publication during a research grant co-funded by the European Social Fund (ESF)—Italian National Operational Programme (NOP) on Research and Innovation 2014–2020 (article 24, clause 3, letter a) of Italian Law n. 240 of 30 December 2010 and of Italian ministerial decree n. 1062 of 10 August 2021, grant number 2021-PON-DM-1062-KS-RIC.

**Institutional Review Board Statement:** Not applicable.

**Informed Consent Statement:** Not applicable.

**Data Availability Statement:** Not applicable.

**Conflicts of Interest:** The authors declare no conflict of interest.

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
