# Peer review of "Appraisal of the Genus Pleurastrum (Chlorophyta) Based on Molecular and Climate Data"

_diversity, doi:10.3390/d15050650_

Round 1

Reviewer 1 Report

Systematics of the Chlamydomonadales group is in dire need of revision, and I was pleased to see that the authors pursued this project of species delineation of Chlorococcum, Macrochloris and Pleurastrum. I think the manuscript as a whole is well written and to the point and recommend it for publication with minor revisions. My comments and suggestions are directed at the discussion.

It's interesting that they change morphology in culture, I think you should emphasise this problem more and clarify that your morphological analyses were done immediately after isolation, such that they represent the organisms' true morphology.

I would like to see some discussion about what evolutionary patterns you would expect to see from the different genes/genomes. Would one expect to see different phylogenetic patterns in the different genes and why?

What is a green alga species? You are recommending systematic revision in your conclusions so I recommend you specify what species concept you follow. I like that you apply both morphology, climate zones and genetics to describe these lineages of green algae, but are you sure you're delineating "species" and not just geographical lineages/subspecies? 

I suggested moderate English revision throughout, but I was between this and minor English revisions. It's very clear and well written throughout, but some different word choices could be made. For example, instead of saying that strains were "got" from XYZ say that strains were "obtained" from XYZ. Instead of "focused" strains say "focal" strains.

Author Response

Systematics of the Chlamydomonadales group is in dire need of revision, and I was pleased to see that the authors pursued this project of species delineation of Chlorococcum, Macrochloris and Pleurastrum. I think the manuscript as a whole is well written and to the point and recommend it for publication with minor revisions. My comments and suggestions are directed at the discussion.

It's interesting that they change morphology in culture, I think you should emphasise this problem more and clarify that your morphological analyses were done immediately after isolation, such that they represent the organisms' true morphology.

Answer: Indeed, first observations with the light microscope were made just after the collection of the Antarctic isolates and the isolates showed characters typical of Pleurastrum. Then, after a period in culture, other observations were made with the light microscope and also with the electron microscope and some features were not observed again. After your suggestion, we added a sentence to explain this in the text. Now, we have decided to add also the electron microscope data in the manuscript.

I would like to see some discussion about what evolutionary patterns you would expect to see from the different genes/genomes. Would one expect to see different phylogenetic patterns in the different genes and why?

Answer: following your suggestion, we have added a part regarding the choice of different genes/genomes in the Discussion.

What is a green alga species? You are recommending systematic revision in your conclusions so I recommend you specify what species concept you follow. I like that you apply both morphology, climate zones and genetics to describe these lineages of green algae, but are you sure you're delineating "species" and not just geographical lineages/subspecies?

Answer: following your suggestion, we have added a part trying to explain this better in the last part of the Discussion.

I suggested moderate English revision throughout, but I was between this and minor English revisions. It's very clear and well written throughout, but some different word choices could be made. For example, instead of saying that strains were "got" from XYZ say that strains were "obtained" from XYZ. Instead of "focused" strains say "focal" strains.

Answer: As suggested, we have replaced “got” with “obtained” throughout the text. About “focused” we have replaced it with “focal” throughout the text, as suggested, and in a case with “focused on”. We have also checked the English throughout the text.

Reviewer 2 Report

The manuscript deals with the taxonomic revision of genera Clorococcum and Pleurastrum. The authors clearly confirm the fact the genus Chlorococcum is polyphyletic by sequencing multiple molecular loci in several strains, and they further presented the results of several DNA-based species delimitation methods. However, taxonomically the manuscript is unsatisfactory and cannot be published in the present form since the taxonomic conclusions are not supported by the presented data.

First, the morphological data presented are poor. In the modern taxonomic studies, molecular genetic data are often overused at the expense of morphology. This manuscript is an exemple of that. For example, I do not see any value in presenting the ITS2 numeric barcode alignment (Fig. 9). Honestly, do the authors really see any practical advantage in presenting such tables, or are they simply using an approach published elsewhere? The Fig. 9 table shows the same information concerning the species relationships, but in much more complicated way (it's the same as if the authors put the entire alignment in the picture). On the other hand, the microphotographs barely show any internal structure of the cells, and their resolution is poor. To get much better results, the algae should be photographed i) during the exponential phase of growth, ii) in the thinnest possible layer of the preparation, and ii) using the DIC contrast. if the authors want to take inspiration from Darienko and Pröschold's work (which they clearly do), I strongly recommend trying to get equally high-quality microalgae photos.

Second, and more importantly, the taxonomic conclusions are incorrect, as they are based on the strain of “Pleurastrum” insigne SAG 30.93, which was obviously overgrown in the culture collection by the Chlorococcum strain. As it is clearly written by Kawasaki et al. (2015), while Sluiman and Gärtner (1990) still observed the formation of pseudoparenchymatous filaments, i.e., one of the important features of the genus Pleurastrum, Kawasaki et al. (2015) did not observe these structures despite cultivation in different media. Therefore, as they concluded correctly, the strain SAG 30.93 is not longer an authentic strain of P. insigne! Comparing the Chodat’s original drawings of Pleurastrum with the figures presented in the manuscript under view, it is again clear the strains presented morphologically match the definition of the Chlorococcum, not Pleurastrum. As I mentioned above, due to the poor quality of microphotographs, it is hard to see in detail the structure of cells and the process of autosporogenesis, but it is highly probable the tetrads and triads illustrated in Fig. 3 (I am quite sure the triads are the tetrads as well, but with one cell hidden) are just autospores. Even if the cells form tetrahedral and sarcionid packets, such feature is typical for the (polyphyletic, as well) genus Tetracystis, not Pleurastrum. The authors state that “the filamentous and sarcinoid forms, typical of the genus Pleurastrum, were observed only just after the isolation of the two strains (data not shown) and then were lost in the laboratory culture conditions”. However, such evidence should be clearly shown as it is essential in the terms of the taxonomic conclusion given! The authors should clearly demonstrate such sarcinoid forms are the real pseudoparenchymatous filaments corresponding to Pleurastrum, and not Tetracystis-like sarcinoid packets or simple clumps of cells or autospores.

In conclusion, I have to recommend rejection of the manuscript since the taxonomic conclusions are not warranted by presented data. Instead, they are erroneously based on the originally authentic but now overgrown strain. If the authors are willing to taxonomically assess the “Chlorococcum” A lineage correctly, they should i) describe the morphology in much more detail (and to present the photos of much better quality), ii) discuss the morphological similarities to all previously recognized morphologically similar taxa (incl. Tetracystis, Cystomonas, Neospongiococcum, Radiosphaera, Heterotetracystis, Spongiococcum, Borodinellopsis, Axilosphaera, and others), to conclude the lineage could be identified with any of previously-described taxa or it represents a new genus, iii) focus more on the taxonomic history of Chlorococcales instead of presenting here irrelevant ITS2 secondary structures, barcodes, and so on.

Author Response

The manuscript deals with the taxonomic revision of genera Clorococcum and Pleurastrum. The authors clearly confirm the fact the genus Chlorococcum is polyphyletic by sequencing multiple molecular loci in several strains, and they further presented the results of several DNA-based species delimitation methods. However, taxonomically the manuscript is unsatisfactory and cannot be published in the present form since the taxonomic conclusions are not supported by the presented data.

ANSWER: Obviously, we do not agree with the opinion of Reviewer 2, since the data are well supported by strong molecular analyses. In particular, four molecular loci were used, with four phylogenetic analysis methods employed for each locus, and the obtained phylogenies have strong statistical supports. Moreover for the ITS2 locus of the ITS region, further analyses were carried out considering both the sequences and the secondary structures: four phylogenetic analysis methods, the CBC and hCBC analysis and four species delimitation statistical methods were also applied in this regard, also considering the most updated molecular methods (e.g. the ASAP method). Our opinion is supported also by the other three reviewers that do not question the quality of our results, but say that “the manuscript is well-written” and “of broad interest” and “very interesting and important for the taxonomy of green algae” and that it can be accepted after minor revision. However, we have tried to address some of the points raised by Reviewer 2.

First, the morphological data presented are poor. In the modern taxonomic studies, molecular genetic data are often overused at the expense of morphology. This manuscript is an exemple of that. For example, I do not see any value in presenting the ITS2 numeric barcode alignment (Fig. 9). Honestly, do the authors really see any practical advantage in presenting such tables, or are they simply using an approach published elsewhere? The Fig. 9 table shows the same information concerning the species relationships, but in much more complicated way (it's the same as if the authors put the entire alignment in the picture). On the other hand, the microphotographs barely show any internal structure of the cells, and their resolution is poor. To get much better results, the algae should be photographed i) during the exponential phase of growth, ii) in the thinnest possible layer of the preparation, and ii) using the DIC contrast. if the authors want to take inspiration from Darienko and Pröschold's work (which they clearly do), I strongly recommend trying to get equally high-quality microalgae photos.

ANSWER: About Figure 9, the utility of present it among the main data is to show better to the readers how the numeric barcode phylogeny was obtained, since it is still less frequently used than other phylogenetic methods, allowing also to other researchers to reproduce the method. Moreover, figures of the numeric barcode alignments are present among the main data in other similar papers on microalgal systematics, including papers published on the journal Diversity. However, if the view of Figure 9 among the main data is such a problem for Reviewer 2 and given the large amount of figures present as main data, it is not a problem for us to move figure 9 to the Supplementary files.

About the question on morphology, it was not the main focus of this paper as declared in the title (i.e. “Appraisal of the genus Pleurastrum (Chlorophyta) based on molecular and climate data”) and given also the special issue to which it was sent: “The phylogenetic Diversity of Cyanobacteria and Algae”. The choice of not focusing too much on morphology was also due to the fact that the investigated group of algae shows simple and overlapping morphologies, often influenced by environmental conditions, and that for the considered strains, in particular, morphology was investigated well in previous papers, while molecular and climate data had been less explored, in particular taken together. However, following the suggestion by Reviewer 2, we decided to add pictures obtained with Scanning and Transmission Electron Microscopy and to discuss the morphological aspects more.

Second, and more importantly, the taxonomic conclusions are incorrect, as they are based on the strain of “Pleurastrum” insigne SAG 30.93, which was obviously overgrown in the culture collection by the Chlorococcum strain. As it is clearly written by Kawasaki et al. (2015), while Sluiman and Gärtner (1990) still observed the formation of pseudoparenchymatous filaments, i.e., one of the important features of the genus Pleurastrum, Kawasaki et al. (2015) did not observe these structures despite cultivation in different media. Therefore, as they concluded correctly, the strain SAG 30.93 is not longer an authentic strain of P. insigne! Comparing the Chodat’s original drawings of Pleurastrum with the figures presented in the manuscript under view, it is again clear the strains presented morphologically match the definition of the Chlorococcum, not Pleurastrum. As I mentioned above, due to the poor quality of microphotographs, it is hard to see in detail the structure of cells and the process of autosporogenesis, but it is highly probable the tetrads and triads illustrated in Fig. 3 (I am quite sure the triads are the tetrads as well, but with one cell hidden) are just autospores. Even if the cells form tetrahedral and sarcionid packets, such feature is typical for the (polyphyletic, as well) genus Tetracystis, not Pleurastrum. The authors state that “the filamentous and sarcinoid forms, typical of the genus Pleurastrum, were observed only just after the isolation of the two strains (data not shown) and then were lost in the laboratory culture conditions”. However, such evidence should be clearly shown as it is essential in the terms of the taxonomic conclusion given! The authors should clearly demonstrate such sarcinoid forms are the real pseudoparenchymatous filaments corresponding to Pleurastrum, and not Tetracystis-like sarcinoid packets or simple clumps of cells or autospores.

ANSWER: In our opinion, to say that strain SAG 30.93 was “obviously overgrown in the culture collection by the Chlorococcum strain” is not correct, as we discuss in the manuscript. In fact, despite their good and wide work, a weakness of the paper by Kawasakii et al. (2015) was that their conclusions were made without considering the type strain of Chlorococcum type species in their analyses and, to say the truth, it is surprising for us that, at the time, the reviewers of that paper did not make this observation. Kawasakii et al. (2015) made a very big and important work by cultivating strain SAG 30.93 in different media, but this anyway does not guarantee the exact reproduction of the environmental conditions that lead to given morphological traits. Indeed, as we say in the manuscript with the support of literature, the loss of some characters, observed from natural algal populations, in strains cultured for a long time has already been reported, even if this aspect was not taken into account by Kawasakii et al. (2015). Thus, to say that our conclusions are incorrect based on a previous finding, of which we have shown the weakness, is wrong. Indeed, science is a dynamic field and perhaps, in the future, other researchers will change our current conclusions, but for the moment our data are supported.

In conclusion, I have to recommend rejection of the manuscript since the taxonomic conclusions are not warranted by presented data. Instead, they are erroneously based on the originally authentic but now overgrown strain. If the authors are willing to taxonomically assess the “Chlorococcum” A lineage correctly, they should i) describe the morphology in much more detail (and to present the photos of much better quality), ii) discuss the morphological similarities to all previously recognized morphologically similar taxa (incl. Tetracystis, Cystomonas, Neospongiococcum, Radiosphaera, Heterotetracystis, Spongiococcum, Borodinellopsis, Axilosphaera, and others), to conclude the lineage could be identified with any of previously-described taxa or it represents a new genus, iii) focus more on the taxonomic history of Chlorococcales instead of presenting here irrelevant ITS2 secondary structures, barcodes, and so on.

Answer: for all the above exposed reasons, we do not agree with Reviewer 2 and, fortunately, we have also the support of the good opinions by the other three reviewers. We hope that, given the heated and somewhere offensive tones of Reviewer 2 (the final sentence, “irrelevant ITS2 secondary structures, barcodes, and so on”, is rather over the top!), she/he would not have conflicts of interest with this manuscript…

Reviewer 3 Report

The manuscript “APPRAISAL OF THE GENUS PLEURASTRUM (CHLOROPHYTA) BASED ON MOLECULAR AND CLIMATE DATA” by Sciiuto et al presents a phylogenetic reconstruction based on molecular markers (i.e. rbcL, tufA and ITS, with a particular focus on ITS2 secondary structures) to more precisely identify the Antarctic isolates (in particular the strains CCCryo 469-16 and CCCryo 470-16, but also other microalga). Their findings demonstrate the relationship between Pleurastrum lineages and specific climate zones. Additionally based on the autor’s analyses, several Chlorococcum species are re-attributed to Pleurastrum, as well as the species Macrochloris rubrioleum, here re-named Pleurastrum rubrioleum comb. nov., to which also the Antarctic isolates belong.

The presented results are of broad interest, therefore, I would recommend the manuscript to be accepted for publication in MDPI “Diversity” after some minor modifications: 

-the images are overall very pixelated (appear to be of low-quality?), which should definitely be improved

-it might be helpful to highlight the new isolates (CCCryo 469-16 and CCCryo 470-16) by color or otherwise within the phylogenetic trees (Fig 4, and others) so that the reader can see the classification of the new strains directly.

In Fig 5, the studied strains CCCryo 469-16 and CCCryo 470-16 cluster within clade A, but the authors write in the text that the strains are divided into two clades (lanes 357-359), or am I misunderstanding something? The authors should separate the new isolates CCCryo 469-16 and CCCryo 470-16 and the other strains for a better understanding.

It is a bit misleading that the authors seem to be focused on the isolates ( CCCryo 469-16 and CCCryo 470-16) at first, but in the manuscript they discuss several isolates at the same time…; Although, I find these information is very important, since allows a greater overall insight, still a better structuring (pointing out) would be desirable.

Author Response

The manuscript “APPRAISAL OF THE GENUS PLEURASTRUM (CHLOROPHYTA) BASED ON MOLECULAR AND CLIMATE DATA” by Sciiuto et al presents a phylogenetic reconstruction based on molecular markers (i.e. rbcL, tufA and ITS, with a particular focus on ITS2 secondary structures) to more precisely identify the Antarctic isolates (in particular the strains CCCryo 469-16 and CCCryo 470-16, but also other microalga). Their findings demonstrate the relationship between Pleurastrum lineages and specific climate zones. Additionally based on the autor’s analyses, several Chlorococcum species are re-attributed to Pleurastrum, as well as the species Macrochloris rubrioleum, here re-named Pleurastrum rubrioleum comb. nov., to which also the Antarctic isolates belong.

The presented results are of broad interest, therefore, I would recommend the manuscript to be accepted for publication in MDPI “Diversity” after some minor modifications: 

-the images are overall very pixelated (appear to be of low-quality?), which should definitely be improved

Answer: we do not understand why the images appear very pixelated to the reviewer, since the images we uploaded in the manuscript are all .jpeg images with a resolution of 600 dpi each. We think that there was a problem in the elaboration of .pdf by the Electronic System, but we hope that it will be not the final version of the manuscript, if it will be accepted for publication.

-it might be helpful to highlight the new isolates (CCCryo 469-16 and CCCryo 470-16) by color or otherwise within the phylogenetic trees (Fig 4, and others) so that the reader can see the classification of the new strains directly.

Answer: After the reviewer observation, we have highlighted the new isolates with a blue asterisk in all the phylogenetic trees.

In Fig 5, the studied strains CCCryo 469-16 and CCCryo 470-16 cluster within clade A, but the authors write in the text that the strains are divided into two clades (lanes 357-359), or am I misunderstanding something? The authors should separate the new isolates CCCryo 469-16 and CCCryo 470-16 and the other strains for a better understanding.

Answer: Indeed, in the paragraph referred to by the reviewer with focused strains we intended the two Antarctic isolates and the strains obtained from International Culture Collections for comparison). Now we have tried to clarify this better in the text.

It is a bit misleading that the authors seem to be focused on the isolates ( CCCryo 469-16 and CCCryo 470-16) at first, but in the manuscript they discuss several isolates at the same time…; Although, I find these information is very important, since allows a greater overall insight, still a better structuring (pointing out) would be desirable.

Answer: After the reviewer observation, we have tried to clarify this better throughout the text.

Reviewer 4 Report

In the reviewed paper very interesting and important for the taxonomy of green algae results are presented. The authors performed the study of two strains from Antarctic, using polyphasic approach (light microscopy, molecular-genetic analysis and estimation of biogeographical distribution). The conclusions of the MS are supported by results. The MS is well edited.

But I have some remarks to the authors.

Lines 21-22, 120-121 and further: Why did you write International Culture Collections with capital letters?

Figures 1, 4-12: Please, improve the quality of the figures, especially the map and phylogenetic trees, because it is impossible to see some inscriptions.

Lines 324-336, 370-377, 396-408, 429-445: In the section “Molecular and phylogenetic results” you listed many strains. This part of the article is very difficult to perceive. Maybe you will try to divide some paragraphs into smaller ones (for example, lines 396-408), or add more details for the strains description (for example, authentic strain, strain from terrestrial habitats and something like this). 

Author Response

In the reviewed paper very interesting and important for the taxonomy of green algae results are presented. The authors performed the study of two strains from Antarctic, using polyphasic approach (light microscopy, molecular-genetic analysis and estimation of biogeographical distribution). The conclusions of the MS are supported by results. The MS is well edited.

But I have some remarks to the authors.

Lines 21-22, 120-121 and further: Why did you write International Culture Collections with capital letters?

Answer: We have seen “International Culture Collections” written both with capital letters and not in different papers. However, it is not a problem for us to use normal letters, so we changed this throughout the text.

Figures 1, 4-12: Please, improve the quality of the figures, especially the map and phylogenetic trees, because it is impossible to see some inscriptions.

Answer: This observation was already made by another reviewer, but indeed we think that this problem is due to the elaboration of .pdf by the Electronic System and we hope that it will be not the final version of the manuscript, if it will be accepted for publication. In fact, all the uploaded figures are .jpeg images with a resolution of 600 dpi each.

Lines 324-336, 370-377, 396-408, 429-445: In the section “Molecular and phylogenetic results” you listed many strains. This part of the article is very difficult to perceive. Maybe you will try to divide some paragraphs into smaller ones (for example, lines 396-408), or add more details for the strains description (for example, authentic strain, strain from terrestrial habitats and something like this).

Answer: We agree with you that this part is very full of strain names and perhaps a bit heavy to read. Nevertheless, it is just the mere description of the trees, with the list of strains as they appear in the clades, and we wrote it just for the completeness of the results. Indeed, it is simpler for the reader just to see the phylogenetic figures. We think that to add further information like “authentic strain”, “strain from terrestrial habitats”, etc. can make this part even heavier. Thus, to follow your suggestion, we could just try to make this part lighter by separating some paragraphs.